# SAMMY-seq reveals early alteration of heterochromatin and deregulation of bivalent genes in Hutchinson-Gilford Progeria Syndrome

Endre Sebestyén [1,8,9], Fabrizia Marullo [2,9], Federica Lucini [3], Cristiano Petrini[1], Andrea Bianchi[2,4], Sara Valsoni [4,5], Ilaria Olivieri [2], Laura Antonelli [5], Francesco Gregoretti [5], Gennaro Oliva[5], Francesco Ferrari [1,6,10 ✉] & Chiara Lanzuolo [3,7,10 ✉]

Hutchinson-Gilford progeria syndrome is a genetic disease caused by an aberrant form of Lamin A resulting in chromatin structure disruption, in particular by interfering with lamina associated domains. Early molecular alterations involved in chromatin remodeling have not been identified thus far. Here, we present SAMMY-seq, a high-throughput sequencing-based method for genome-wide characterization of heterochromatin dynamics. Using SAMMY-seq, we detect early stage alterations of heterochromatin structure in progeria primary fibroblasts. These structural changes do not disrupt the distribution of H3K9me3 in early passage cells, thus suggesting that chromatin rearrangements precede H3K9me3 alterations described at later passages. On the other hand, we observe an interplay between changes in chromatin accessibility and Polycomb regulation, with site-specific H3K27me3 variations and transcriptional dysregulation of bivalent genes. We conclude that the correct assembly of lamina associated domains is functionally connected to the Polycomb repression and rapidly lost in early molecular events of progeria pathogenesis.

[1] IFOM, The FIRC Institute of Molecular Oncology, Milan, Italy. [2] Institute of Cell Biology and Neurobiology, National Research Council, Rome, Italy. [3] Istituto Nazionale Genetica Molecolare "Romeo ed Enrica Invernizzi", Milan, Italy. [4] IRCCS Santa Lucia Foundation, Rome, Italy. [5] Institute for High Performance Computing and Networking, National Research Council, Naples, Italy. [6] Institute of Molecular Genetics, National Research Council, Pavia, Italy. [7] Institute of Biomedical Technologies, National Research Council, Milan, Italy. [8] Present address: 1st Department of Pathology and Experimental Cancer Research, Semmelweis University, Budapest, Hungary. [9] These authors contributed equally: Endre Sebestyén, Fabrizia Marullo. [10] These authors jointly supervised this work: Francesco Ferrari, Chiara Lanzuolo. ✉email: francesco.ferrari@ifom.eu; chiara.lanzuolo@cnr.it

Electron microscopy imaging of eukaryotic nuclei shows chromatin compartments with different levels of compaction, known as euchromatin and heterochromatin (reviewed in refs. [1,2]). The more accessible and less condensed euchromatin is generally enriched in expressed genes. Instead, heterochromatin contains highly condensed DNA, including pericentromeric and telomeric regions[3–5], and genomic regions with unique packaging properties maintained by the Polycomb-group proteins (PcG)[6]. These proteins are developmentally regulated factors acting as parts of multimeric complexes named Polycomb Repressive Complex 1 and 2 (PRC1 and PRC2)[7].

Another key player of chromatin organization is the nuclear lamina (NL). NL is mainly composed of A- and B-type lamins, intermediate filaments of 3.5 nm thickness which form a dense network at the nuclear periphery[8,9]. NL preferentially interacts with the genome at specific regions called Lamina-Associated Domains (LADs), with sizes ranging from 100 kb to 10 Mb[10]. LADs are enriched in H3K9me2 and H3K9me3 histone modifications that are typical of inactive heterochromatic regions. LAD borders are marked by the PRC2-dependent H3K27me3 histone mark[11–13], which is characteristic of inactive PcG-regulated chromatin regions. The ensemble of lamins, chromatin marks, and PcG factors around LADs creates a repressive environment[14,15], with heterochromatin and PcG target regions adjacent to each other[16,17]. In line with these observations, we previously demonstrated that Lamin A functionally interacts with PcG[18–20], as later also reported by others[17,21–23].

In physiological conditions, the association or detachment of LADs from the NL are related to the spatiotemporal regulation of gene expression in cell differentiation processes[13,24,25]. The crucial function of such dynamics is attested by an entire class of genetic diseases, called laminopathies, where specific components of the NL are altered[13,22,26]. Among them, Hutchinson–Gilford Progeria Syndrome (HGPS, OMIM 176670) is caused by a de novo point mutation in the *LMNA* gene[27], resulting in a truncated splicing mutant form of Lamin A protein, known as progerin. The disease phenotype is characterized by the early onset of several symptoms, resulting in severe premature aging in pediatric patients. At the cellular level, the accumulation of progerin induces progressive alterations of nuclear shape and cellular senescence[28]. Notably, inhibition of PcG functions can trigger "nuclear blebbing", an alteration of the nuclear shape typically observed in Lamin A-deficient cells[29], suggesting that chromatin architecture and nuclear shape strictly depend on each other. Indeed, LADs organization and PcG-dependent H3K27me3 mark are both altered in HGPS[30,31]. Extensive disruption of chromatin structure has been reported in late-passage HGPS fibroblasts by high-throughput genome-wide chromosome conformation capture (Hi-C)[30,32], yet no alterations could be detected in early-passage cells using this experimental technique[30]. This is a crucial distinction, as early-passage HGPS cells are commonly considered a model to identify early molecular alterations in the disease.

In recent years, the widespread adoption of experimental techniques based on high-throughput sequencing (NGS) has been instrumental in advancing the knowledge of chromatin structure and function[33,34]. Several of these techniques were originally designed to map active chromatin regions, including ChIP-seq[35], ATAC-seq[36], DNase-seq[37], and MNase-seq[38]. On the other hand, a limited set of options is available for genome-wide mapping of heterochromatin and LADs. The most commonly used ones include ChIP-seq or DamID-seq[39] targeting NL components or ChIP-seq for the heterochromatin-associated histone marks (H3K9 methylation). Such techniques suffer major limitations as DamID-seq relies on the exogenous expression of a transgene and it cannot be applied to primary cells or tissues. Instead, ChIP-seq relies on crosslinking and antibodies, which are sources of technical biases and potential issues of sensitivity or cross-reactivity, respectively. More recently proposed high-throughput sequencing methods for genome-wide mapping of LADs and heterochromatin, such as gradient-seq[40,41] and protect-seq[42], also rely on crosslinking.

Here, we present a high-throughput sequencing-based method to map lamina-associated heterochromatin regions. We named our technology SAMMY-seq (Sequential Analysis of Macro-Molecules accessibilitY), as a tribute to Sammy Basso, the founder of the Italian Progeria Association, a passionate and inspiring advocate of research on laminopathies. SAMMY-seq relies on the sequential isolation and sequencing of multiple chromatin fractions, enriched for differences in accessibility. Here we show that our method is robust, fast, easy-to-perform, it does not rely on crosslinking or antibodies, and it can be applied to primary cells. In healthy human fibroblasts, we can reliably identify genomic regions with heterochromatin marks (H3K9me3) and LADs. Of note, other high-throughput sequencing methods relying on high salt buffers to extract specific chromatin fractions have been proposed in the literature, including "salt fractions profiling"[43] and HRS-seq[44]. However, they adopt different conditions, enzymes, experimental protocol steps (Supplementary Table 1), and cellular models. Moreover, they have been primarily adopted to analyze accessible euchromatin regions, whereas SAMMY-seq allows the study of the condensed heterochromatin domains, thus they are not directly comparable to our technology.

Using SAMMY-seq in early-passage HGPS fibroblasts, we found patient-specific alterations of heterochromatin architecture, but this is not accompanied by H3K9me3 changes or transcriptional activation of silent LAD. Interestingly, we also identified a global decrease of the PRC2-dependent H3K27me3 mark deposition, suggesting an interplay between differentially compacted domains and PcG regulation. These Polycomb-dependent alterations affect the transcriptional regulation of bivalent genes, more susceptible to H3K27me3 variations[18,45]. Globally, our findings indicate that chromatin structural changes are an early event in HGPS nuclear remodeling and interfere with proper PcG control.

## Results

**SAMMY-seq maps lamina-associated heterochromatic regions.** We developed the SAMMY-seq method for genome-wide mapping of chromatin fractions separated by accessibility. The method is based on the sequential extraction of distinct nuclear fractions[19,46,47] containing: soluble proteins (S1 fraction); DNase-treated chromatin, i.e., the supernatant obtained after DNase treatment (S2 fraction); DNase-resistant chromatin extracted with high salt buffer (S3 fraction); and the most condensed and insoluble portion of chromatin, extracted with urea buffer that solubilizes the remaining proteins and membranes (S4 fraction) (Fig. 1a and Supplementary Fig. 1a). We further adapted the method to leverage high-throughput DNA sequencing for genome-wide mapping of the distinct chromatin fractions (see "Methods" section).

We first applied SAMMY-seq to three independent skin primary fibroblast cell lines, originating from three different healthy individuals (control samples). The average number of sequenced reads was 72 million for S2 fraction, 62 million for S3 fraction, and 65 million for S4 fraction (Fig. 1b and Supplementary Fig. 1b, Supplementary Table 2). On average 58% (S2), 54% (S3), and 52% (S4) of the genome were covered by at least one read (Supplementary Fig. 1c). The pairwise Spearman correlation between control samples (Supplementary Fig. 1d) generally increases when summarizing read counts over larger genomic

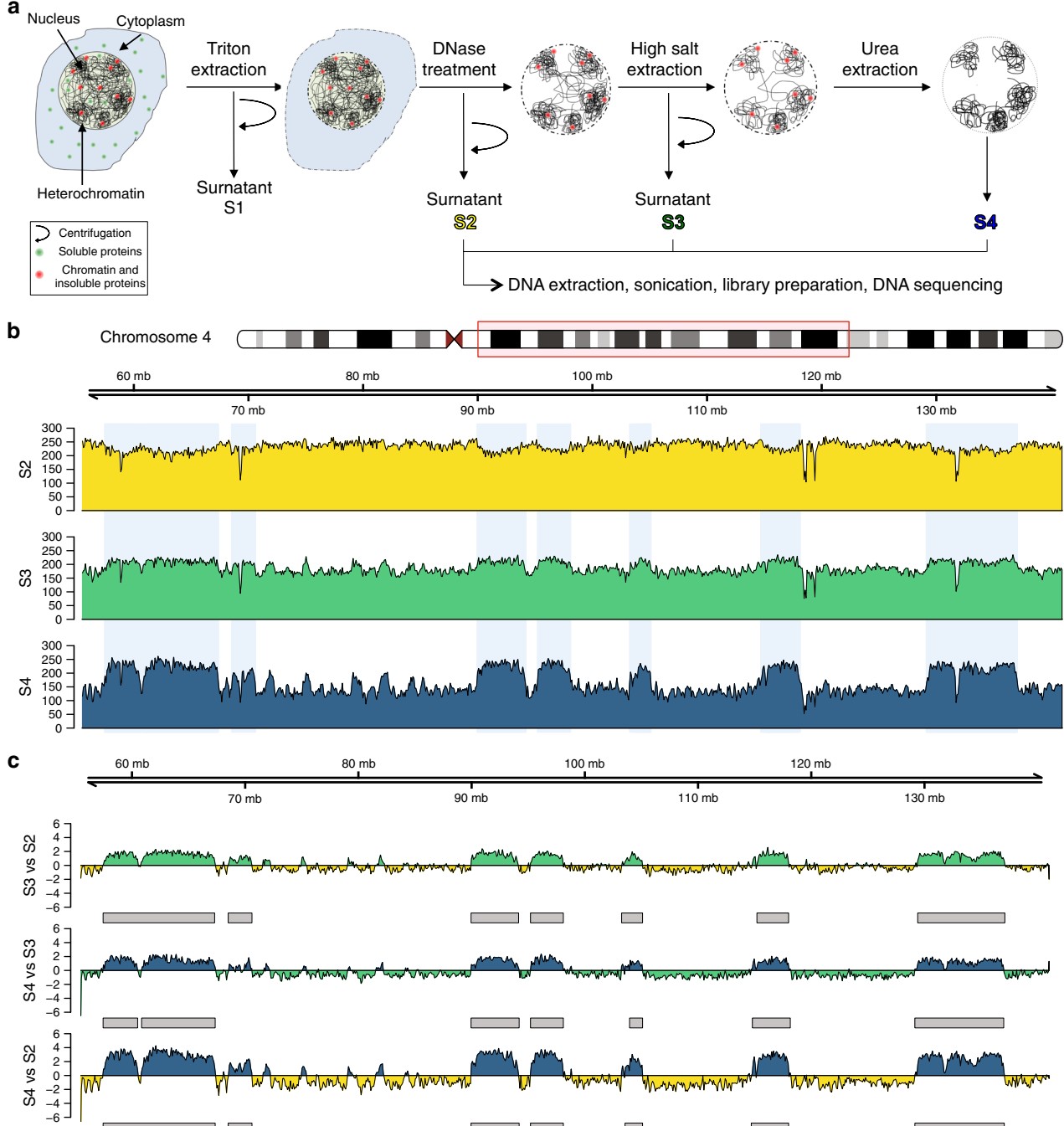

**Fig. 1 SAMMY-seq isolates specific DNA regions in fibroblast samples. a** Schematic representation of SAMMY-seq. Chromatin fractions are sequentially isolated after DNase treatment, high salt, and urea buffers extractions. The associated genomic DNA is extracted, sonicated, and processed for high-throughput sequencing. **b** Distribution of SAMMY-seq reads along a representative region (85 Mb region in chr4:55480346-141005766). Library size normalized read counts over 10 kb genomic bins are shown for each sequenced fraction of a control fibroblast sample (CTRL002) at passage 20.
**c** Differential reads distribution across pairwise comparisons of SAMMY-seq fractions in a control fibroblast sample (CTRL002) along the genomic region in **b**. Less accessible fractions are compared to more accessible ones used as reference (S3 vs S2; S4 vs S3; S4 vs S2). Regions of signal enrichment (green or blue) or depletion (yellow or green) over the reference samples are shown. The smoothed differential signal is calculated with SPP, and significantly enriched regions (SAMMY-seq domains) are called with EDD and reported as gray boxes under the enrichment signal track.

bins and reaches a plateau for S3 and S4 fractions (Supplementary Fig. 1e). Around 1 Mb resolution, the mean Spearman correlation between control samples is 0.27 for the S2 fraction, 0.92 for S3, and 0.79 for S4. Overall, these results show that S3 and S4 fractions show high similarity whereas the S2 fraction is more variable across biological replicates of control samples (Supplementary Fig. 1d).

In line with this analysis, the visual inspection of read coverage profiles shows megabase-scale "bumps" in S3 and, even more prominently, in S4 fractions (Fig. 1b). It is worth remarking that the S2 profile is not comparable to the standard DNase-seq profile due to substantial differences in the digestion conditions applied here (see "Methods" section). To detect relative enrichment of specific genomic regions between chromatin fractions, we applied

the Enriched Domain Detector (EDD)[48], an algorithm designed to identify broad regions of enrichment in high-throughput sequencing data. We identified differentially enriched genomic regions by pairwise comparisons of less accessible vs more accessible fractions, the latter used as a baseline in the comparison: i.e., S3 vs S2, S4 vs S3, and S4 vs S2 comparisons (Fig. 1c). We found a well-defined set of regions (SAMMY-seq domains) consistently enriched in the less accessible fractions. The S4 vs S2 comparison is the most consistent with 70.18% conservation between sample pairs on average (Supplementary Fig. 1f). These results are robust with respect to sequencing depth, as we still identify on average 82.09% of the original SAMMY-seq domains after down-sampling to 50% of the sequencing reads (Supplementary Fig. 1g). Further down-sampling to 25% of reads still led to an average of 61.92 % of conserved domains, with S4 vs S2 still showing the most consistent results (Supplementary Fig. 1g). SAMMY-seq domains have an average size of 2.26 Mb and cover on average 527 Mb (Supplementary Fig. 1h, i and Supplementary Table 3). Based on the observations above, we mostly focused on S4 vs S2 comparison in the subsequent analyses, although similar conclusions can be obtained with other comparisons as well.

To functionally characterize the SAMMY-seq enriched regions, we compared the genome-wide read coverage with reference chromatin mark profiles for the same cell type. Using data from the Roadmap Epigenomics consortium[49] and other public data sets[30,48,50–52], we noticed that the SAMMY-seq signal (S4 vs S2 enrichment) is inversely correlated with open chromatin marks (ATAC-seq and DNase-seq) and is positively correlated with Lamin A/C and B1 ChIP-seq signal (Fig. 2a). We confirmed this is a consistent genome-wide pattern using StereoGene kernel correlation[53], a method for the unbiased comparison of different types of chromatin marks (Fig. 2b and Supplementary Fig. 2a). Despite some inter-dataset variability across a comprehensive set of independent Lamin ChIP-seq profiles (Lamin A/C and Lamin B1) for human fibroblasts[30,48,50–52], both Lamin A/C and Lamin B1 show similar correlation values (Supplementary Fig. 2a), thus only their average is summarized in (Fig. 2b). Similarly, when considering histone modifications, we observed a positive association with heterochromatin (H3K9me3) and an inverse correlation with active chromatin (H3K4me1) as well as Polycomb-regulated chromatin (H3K27me3) (Fig. 2a, b). The latter observation is compatible with previous reports that H3K27me3 is located at the border of heterochromatic LAD regions[11–13,17]. However, this should not be interpreted as a co-localization of H3K27me3 and H3K4me1. They are simply concordant in being absent from H3K9me3 enriched SAMMY-seq domains on a large scale. Indeed, a closer inspection of histone mark profiles shows that H3K4me1 peaks are located at positions of relatively lower H3K27me3 enrichment, and vice versa (Fig. 2a). Overall, these observations suggest a similarity between SAMMY-seq and epigenomic profiles for heterochromatin or lamina association.

To further quantify this association, we compared SAMMY-seq domains with Lamin A/C LADs, thus observing an average 0.49 Jaccard Index overlap, when considering all samples and fractions comparisons (Supplementary Table 4). Each of these pairwise comparisons has a significantly high overlap, as assessed by randomizing either LADs or SAMMY-seq domains (Fig. 2c and Supplementary Fig. 2b). On average 73% of SAMMY-seq domains (S4 vs S2 fractions comparison) overlap with Lamin A/C LADs (Supplementary Table 4). Instead, in the complementary comparison, we observed that 62% of Lamin A/C LADs overlap with SAMMY-seq domains (S4 vs S2) (Supplementary Fig. 2c). This observation confirms that LADs cover a larger fraction of the genome compared to SAMMY-seq domains. The

first percentage would be the SAMMY-seq precision in identifying LADs, whereas the latter would be the sensitivity in calling LADs. In a further analysis, we found that the sensitivity in calling regions that are both heterochromatic (H3K9me3) and associated with the lamina (LADs) is actually higher (75% on average for S4 vs S2 comparison) than the sensitivity observed when considering only one of the two markers independently (Supplementary Fig. 2c). Overall, these results confirm SAMMY-seq preferential enrichment for the most compact heterochromatin, such as the lamina-associated heterochromatic regions.

It must be noted that SAMMY-seq overcomes specific limitations of other high-throughput sequencing-based methods for genome-wide mapping of LADs and heterochromatin, thus allowing a wide range of applications (Supplementary Table 5). Most notably, SAMMY-seq does not rely on crosslinking, antibodies, or the overexpression of a transgene, thus reducing the sources of biases and extending its applicability. A key feature of SAMMY-seq, that will allow its adoption to a wide range of applications, is the possibility to analyze a small number of cells. To this concern, as a proof of concept, we performed a scale-down experiment by applying SAMMY-seq on a control fibroblast cell line (CTRL004) starting from 250k, 50k, or 10k cells. To further prove the robustness of the technique, the experiment was performed on two replicates by two independent experimentalists. The SAMMY-seq profiles obtained in the scale-down experiment confirm the association of S4 fraction with heterochromatin and lamina (Supplementary Fig. 3a) as well as the good reproducibility of the profile across replicates and scale-down tests. These patterns are confirmed by the genome-wide pairwise correlation between SAMMY-seq samples, as well as their correlation with H3K9me3, and negative correlation with H3K4me1 and H3K27me3 (Supplementary Fig. 3b). The pairwise overlap of scale-down SAMMY-seq domains (Jaccard Index) also confirms the robustness of results (Supplementary Fig. 3c). Globally, these results indicate that SAMMY-seq is a versatile technique that can reliably identify chromatin regions separated on their accessibility.

**Chromatin changes in early-passage progeria fibroblasts.** We applied SAMMY-seq on early-passage HGPS primary fibroblasts, generally used as a cellular model to study early molecular alterations in progeria[30,54]. Indeed, progerin is known to be resistant to protein degradation within cells[55] and to accumulate during cell passaging (Supplementary Fig. 4a). According to the literature, primary HGPS fibroblasts entering senescence show alterations in DNA interactions with Lamin A/C[30]. However, Hi-C experiments for genome-wide mapping of 3D chromatin architecture could not detect alterations in the pairwise contact frequency between genomic loci in early-passage cells. These were evident only at later-passage cells, representing more advanced stages of senescence when cells exhibit severe nuclear blebs and inhibited proliferation[30].

We confirmed that early-passage HGPS fibroblasts do not show senescence-associated features such as beta-galactosidase positivity (Supplementary Fig. 4b), reduction in the proliferation rate (Supplementary Fig. 4c), or changes in nuclear area and morphology (Supplementary Fig. 4d–f). We applied SAMMY-seq to investigate chromatin changes in early-passage skin fibroblasts from 3 independent HGPS patients (passage number from 10 to 12) (Supplementary Fig. 1b, c). SAMMY-seq domains (S4 vs S2) appeared more variable in number and dimension across samples with respect to controls (Supplementary Fig. 1h–i), and scattered and with a generally lower signal-to-noise ratio (Fig. 3a) in HGPS, as opposed to the more consistent overlap across control replicates. This would be compatible with a general loss of

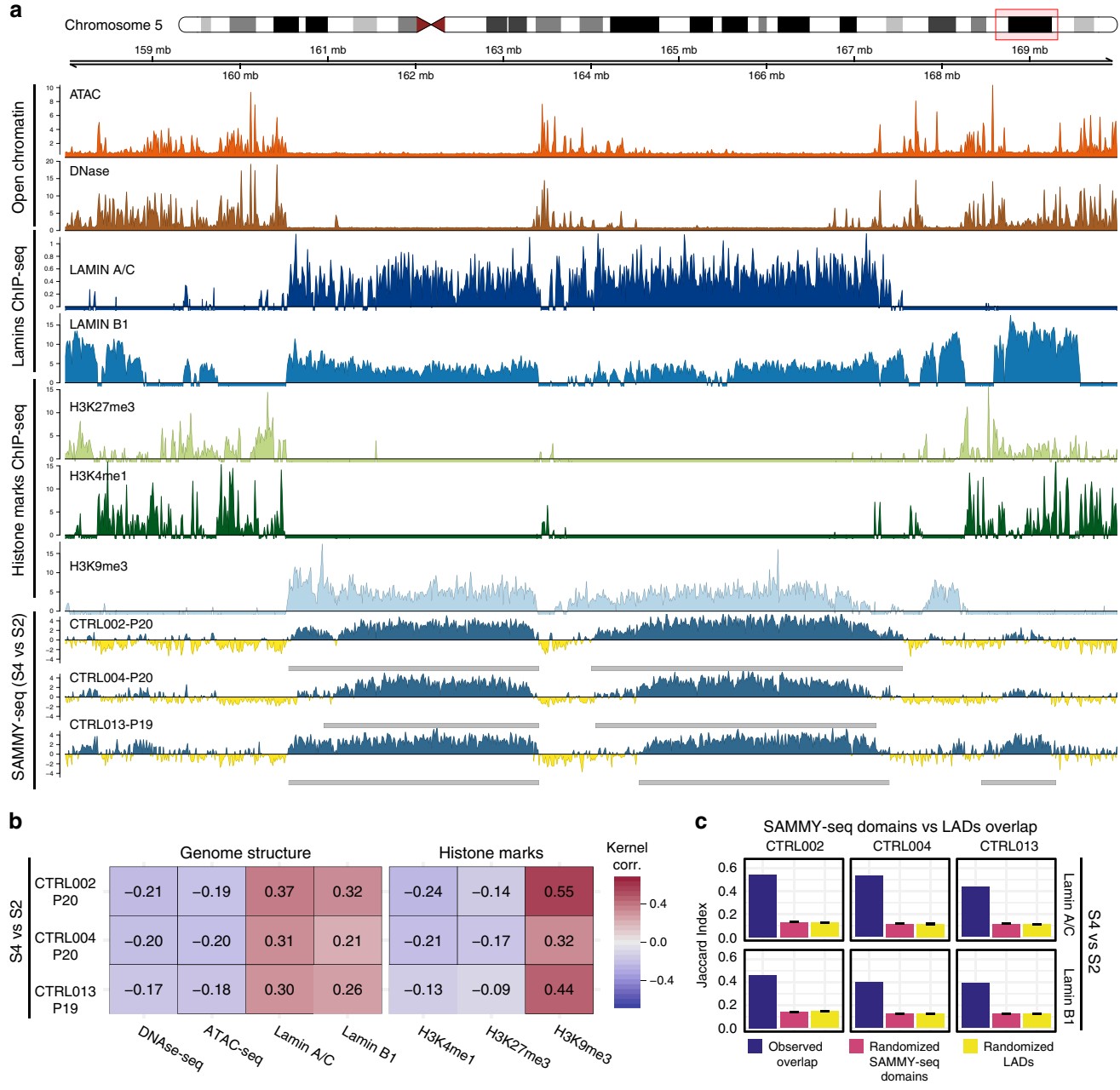

**Fig. 2 SAMMY-seq preferentially isolates heterochromatic domains. a** Visualization of SAMMY-seq enrichment signal along with multiple chromatin marks on a representative region (12 Mb region in chr5:158000000-170000000). From top to bottom: tracks for open chromatin (ATAC-seq and DNase-seq – orange and brown); association to the lamina (Lamin A/C and Lamin B1 ChIP-seq – dark blue and blue); ChIP-seq for histone marks associated to PcG regulation (H3K27me3 – light green), active chromatin (H3K4me1 – dark green) or heterochromatin (H3K9me3 – light blue); SAMMY-seq enrichment signal (S4 vs S2) in 3 control samples at indicated passages (p19-20) (blue or yellow colored track for enrichment or depletion, respectively, over the S2 reference baseline). Gray boxes under each SAMMY-seq track show the SAMMY-seq domains. For ChIP-seq data, the (ChIP – input) reads distribution was computed with SPP package and we are limiting the y axis range to zero as a minimum value. See the "Methods" section for details of each type of signal processing. **b** Genome-wide kernel correlation (StereoGene) for SAMMY-seq of all control samples at indicated passages (p19-20) (S4 vs S2 enrichment) compared to ATAC-seq, DNase-seq, and ChIP-seq (Lamin A/C, Lamin B1, H3K27me3, H3K4me1, and H3K9me3). Here we are presenting a subset of data reported in Supplementary Fig. 2a containing a more extended and detailed comparison. For Lamin A/C and Lamin B1, we are reporting the average correlation across multiple data sets (see Supplementary Fig. 2a for individual data sets). **c** Overlap (Jaccard Index - JI) of three biologically independent (labels on top) control samples SAMMY-seq domains (S4 vs S2) with Lamin-Associated Domains (LADs) (Lamin A/C or Lamin B1 ChIP-seq enrichment domains). For Lamin A/C and Lamin B1, we used a consensus set of regions for each (see "Methods" section). The observed JI (blue bar) is compared to mean JI across 10,000 randomizations of SAMMY-seq domains (pink bar) or LADs (yellow bar) positions along the genome to compute empirical p-values for the probability of the observed JI being larger than the random expectation. All one-tail empirical p-values were significant (p < 0.0001). The whiskers over the randomized values show the mean ± 2 SEM interval.

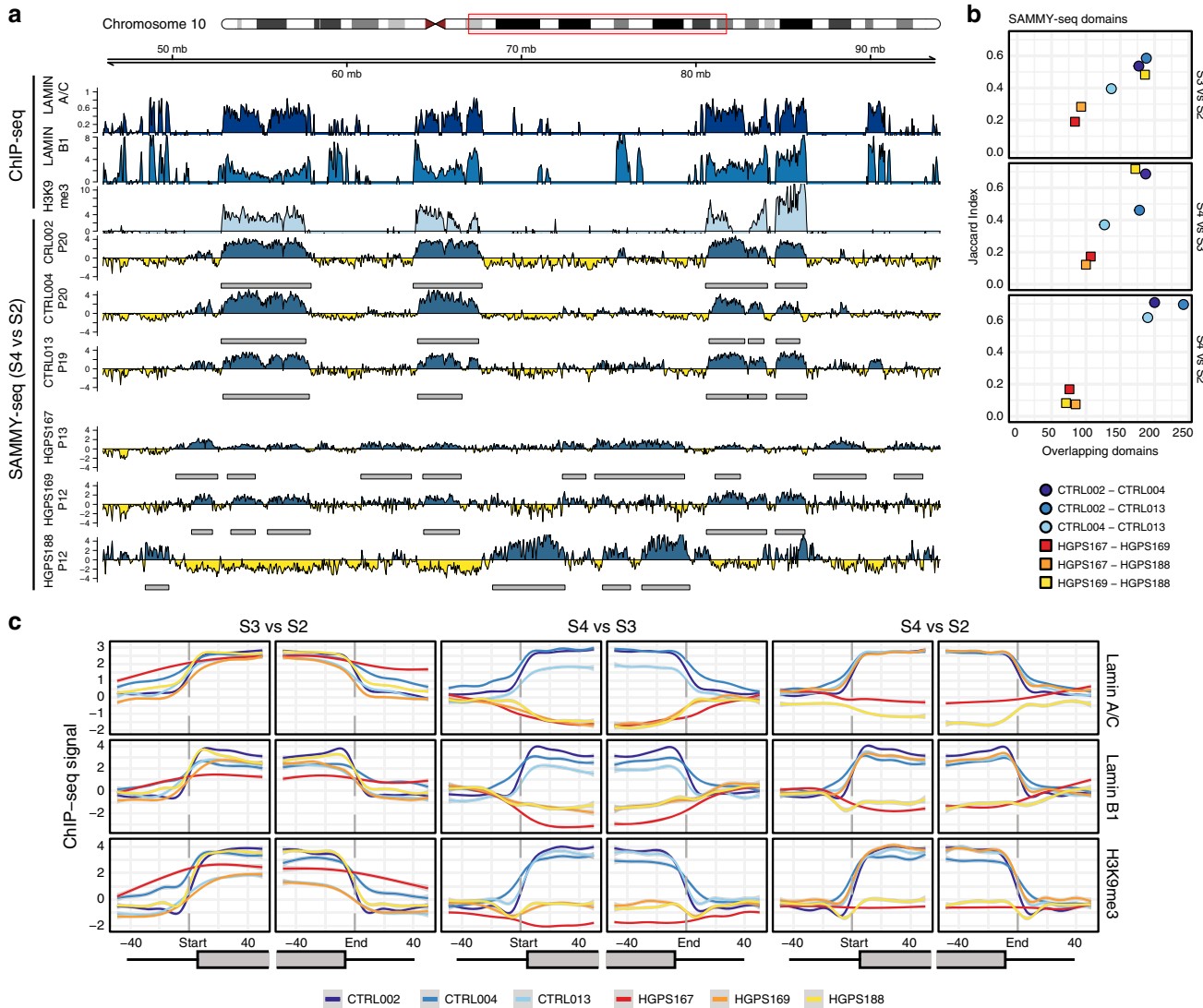

**Fig. 3 HGPS fibroblasts show early changes in chromatin accessibility. a** Genomic tracks for SAMMY-seq in three control and three HGPS samples at indicated passages (p12-20) along with chromatin marks in a representative region (48 Mb region in chr10:46000000-94000000). From top to bottom: tracks for the association to lamina (Lamin A/C and Lamin B1 ChIP-seq – dark blue and blue); heterochromatin (H3K9me3 ChIP-seq – light blue); SAMMY-seq enrichment signal (S4 vs S2) in three control and three HGPS samples (blue or yellow-colored track for enrichment or depletion, respectively, over the S2 reference baseline). Gray boxes under each SAMMY-seq track show the SAMMY-seq domains. For ChIP-seq data, the (ChIP – input) reads distribution was computed with SPP package and we are limiting the y axis range to zero as a minimum value. **b** Pairwise overlap of SAMMY-seq domains (S3 vs S2, S4 vs S3, S4 vs S2) between control or HGPS sample pairs (JI on y axis, number of overlapping regions on x axis). Upper-tail Fisher test p-value < 0.01 for all overlaps, except for HGPS S4 vs S2 pairwise overlaps. **c** Smoothed average ChIP-seq enrichment signal for chromatin marks around SAMMY-seq domain borders. ChIP-seq signal for Lamin A/C (upper row), Lamin B1 (middle row), and H3K9me3 (bottom row) is reported around the SAMMY-seq domain borders start (left side plots) or end (right side plots) for domains detected in each control and progeria sample. Results for each set of SAMMY-seq enrichment domains are reported (S3 vs S2 on the left, S4 vs S3 center, S4 vs S2 on the right) using a ±50 bins window (10 kb bin size) centered on the start or end domain border positions (vertical dashed gray line). The smoothed average is obtained by GAM (see "Methods" section) and the gray shaded area under each line shows the 95% confidence interval for the fitted GAM.

chromatin organization following the lamina structural alteration, thus resulting in a more random distribution of genomic regions among the different chromatin fractions. Indeed, SAMMY-seq domains are largely overlapping among control samples (average Jaccard Index 0.67) whereas SAMMY-seq domains are more variable in progeria samples, most notably in the S4 vs S2 comparison (average Jaccard Index 0.11) (Fig. 3b and Supplementary Table 6). Moreover, SAMMY-seq domains in control samples fall within regions that are normally enriched for lamins and H3K9me3, whereas this pattern is lost in HGPS SAMMY-seq

samples (Fig. 3a). We confirmed this is a genome-wide trend by analyzing chromatin marks at SAMMY-seq domain borders (Fig. 3c). SAMMY-seq domains from control samples (S4 vs S2, S4 vs S3, S3 vs S2) correspond to regions normally enriched in Lamin A/C, Lamin B1 and H3K9me3 ChIP-seq reads, as opposed to flanking regions outside of domain borders. Progeria SAMMY-seq domains (S3 vs S2) are similar to controls. However, when considering a more compacted S4 fraction, the association to reference controls for LADs and heterochromatin marks is lost in 2 out of 3 patients for the S4 vs S2, and 3 out of 3 patients for the

S4 vs S3 comparisons, respectively (Fig. 3c). Here, both the lamins and the H3K9me3 signals are flattened out suggesting no specific association. Overall, these observations confirm a high inter-individual variability among HGPS patients and suggest that early-passage HGPS samples loose the pattern normally observed in controls.

We also performed SAMMY-seq on late-passage control (CTRL004) and HGPS (HGPS167) primary fibroblasts maintained in culture up to passage 27 and 20, respectively, i.e., about 3 months from the early-passage cells (Supplementary Fig. 5). Independently passaged HGPS biological replicates showed a pattern still different from the late-passage control sample in the overlap of SAMMY domains (Supplementary Fig. 5a, b). Nevertheless, we noticed a partial recovery of the control pattern in the S4 vs S2 comparison, as confirmed by genome-wide correlation (Supplementary Fig. 5c) and association with heterochromatin marks (Supplementary Fig. 5d). Of note, the S4 vs S3 comparison still highlighted consistent differences between the control and HGPS profiles (Supplementary Fig. 5e–h), suggesting that SAMMY fractions give rise to complementary information in chromatin analysis.

The seemingly partial recovery of the normal fibroblasts pattern in late-passage HGPS cells can be explained by the progerin-dependent inhibition of the cell cycle and DNA damage accumulation[56–58], generally resulting in more apoptosis in late-passage HGPS cells. Thus, prolonged culture may actually counter-select the cells with more abundant progerin. Moreover, the permeabilization of cell and nuclear membranes in the initial step of SAMMY-seq protocol would result in disrupting apoptotic bodies and discard their content, thus further under-representing the more damaged cells. It is worth noting that other chromatin analysis methods starting with a fixation step (e.g., crosslinking) would not equally clear apoptotic bodies, thus confounding the results.

**H3K9me3 patterns do not change in early-passage HGPS cells**. To further investigate the epigenetic regulation in early-passage HGPS fibroblasts, we examined H3K9me3 distribution by ChIP-seq on the same set of control and HGPS fibroblasts analyzed by SAMMY-seq. H3K9me3 profiles were mostly unchanged in early-passage HGPS, despite the chromatin remodeling detected by SAMMY-seq (Fig. 4a). Quantitative analyses confirmed that H3K9me3 enriched regions are largely conserved between controls and HGPS cells, with overlaps ranging between 69 and 80% (Fig. 4b). Only controls exhibit a correlation between SAMMY-seq and H3K9me3, as indicated by the analysis of H3K9me3 pattern at SAMMY-seq domain borders (Supplementary Fig. 6a) as well as by the Jaccard index analysis (Fig. 4c and Supplementary Fig. 6b).

In line with genomic data, at the cellular level, early-passage HGPS cells and controls showed comparable H3K9me3 global levels measured by western blot (Supplementary Fig. 6c, d). Heterochromatin foci were detected in both controls and early-passage HGPS fibroblasts by immunofluorescence analysis with H3K9me3 antibody, revealing high variability across samples (Fig. 4d) but without significant differences in their number per cell (Supplementary Fig. 6e). Instead, their intranuclear localization showed a shift towards the nuclear center in HGPS (Fig. 4d, e), supporting the hypothesis of heterochromatin detachment from the nuclear periphery (Fig. 3). Notably, as previously reported[31], later passages (p20) HGPS fibroblasts showed an aberrant pattern of H3K9me3 foci (Fig. 4d), further indicating that early-passage chromatin remodeling in HGPS precedes H3K9me3 loss.

**Chromatin changes do not affect the expression of LAD genes**. Using RNA-seq on the same set of control and HGPS fibroblasts,

we assessed the expression at the gene or transcript-level detecting differential expression of 257 genes or 256 transcripts, respectively (Supplementary Fig. 7a) (see "Methods" section). Thus, early stages of senescence in HGPS cells do not cause large-scale changes in the transcriptome. To clarify whether chromatin structural changes detected with SAMMY-seq have an effect on transcription, we examined the expression of protein-coding genes located in SAMMY-seq domains. As expected[16], genes within SAMMY-seq domains (S4 vs S2) have lower expression levels than those outside in all control samples (Supplementary Fig. 7b). More specifically, in the genomic regions around SAMMY-seq domain borders, we noticed a clear pattern with a transition from lower expression within the domain to higher expression outside in all of the control samples (Supplementary Fig. 7c). A similar pattern was observed in HGPS samples when considering the same genomic regions (Supplementary Fig. 7b, c). This suggests that heterochromatin detachment from NL seen in HGPS (Figs. 3 and 4e) was not sufficient to trigger gene expression deregulation inside SAMMY-seq domains, where H3K9me3 is still present and largely unchanged as shown above (Fig. 4a, b). The transcriptional activity within H3K9me3 domains of each sample is repressed in HGPS as well as in controls, thus further confirming that epigenetic silencing is still preserved in these regions (Supplementary Fig. 7d).

**HGPS chromatin remodeling affects PcG bivalent genes**. As we noticed that each HGPS sample has unique SAMMY-seq domains (Fig. 3), we compared individual HGPS expression profiles against the group of three controls to account for sample-specific patterns (see "Methods" section). To verify which biological functions are affected, we performed a pathway enrichment analysis using the MSigDB database[59] and selected the pathways commonly deregulated in all HGPS samples (Fig. 5a). Many of them are related to stem cells and cancer, in line with the hypothesis that accelerated aging affects the stem cell niche[60]. Interestingly, we also found several pathways related to PcG regulation (H3K27me3 and PRC2) (Fig. 5a), in line with previous studies suggesting crosstalk between lamina and PcG[17–19,21–23]. To further investigate PcG role in early-passage HGPS fibroblasts, we examined the total amounts of PRC1 (Bmi1 subunit) and PRC2 (Ezh2 subunit) at the protein level (Supplementary Fig. 8a, b), as well as H3K27me3 (Supplementary Fig. 8c, d). None of them showed significant differences between control and HGPS samples, despite some degree of inter-individual variability. We also checked the intranuclear architecture of Ezh2 (Supplementary Fig. 8e, f) and H3K27me3 foci (Supplementary Fig. 8g, h) and we found that their number does not change at early-passage HGPS cells.

Finally, we performed ChIP-seq to dissect H3K27me3 distribution at the genome-wide level. By visual inspection, we noticed that the main enrichment peaks seem to be present in both controls and HGPS samples, but the H3K27me3 signal was slightly spreading over flanking regions in HGPS (Fig. 5b). To quantify this pattern, we considered the annotated transcription start site (TSS) for protein-coding genes where H3K27me3 was detected in controls. We examined the H3K27me3 average profile around these TSS regions, where we detected generally lower enrichment in HGPS samples, as well as a less marked relative decrease in TSS distal vs proximal positions, yet not statistically significant due to variability across patients (Supplementary Fig. 9a, b). This observation resembles the findings of H3K27me3 spreading described in another laminopathy model[18]. The progeria-associated differences in the H3K27me3 profile were also confirmed by multi-dimensional scaling of ChIP-seq data (Fig. 5c), showing that control samples are close to each

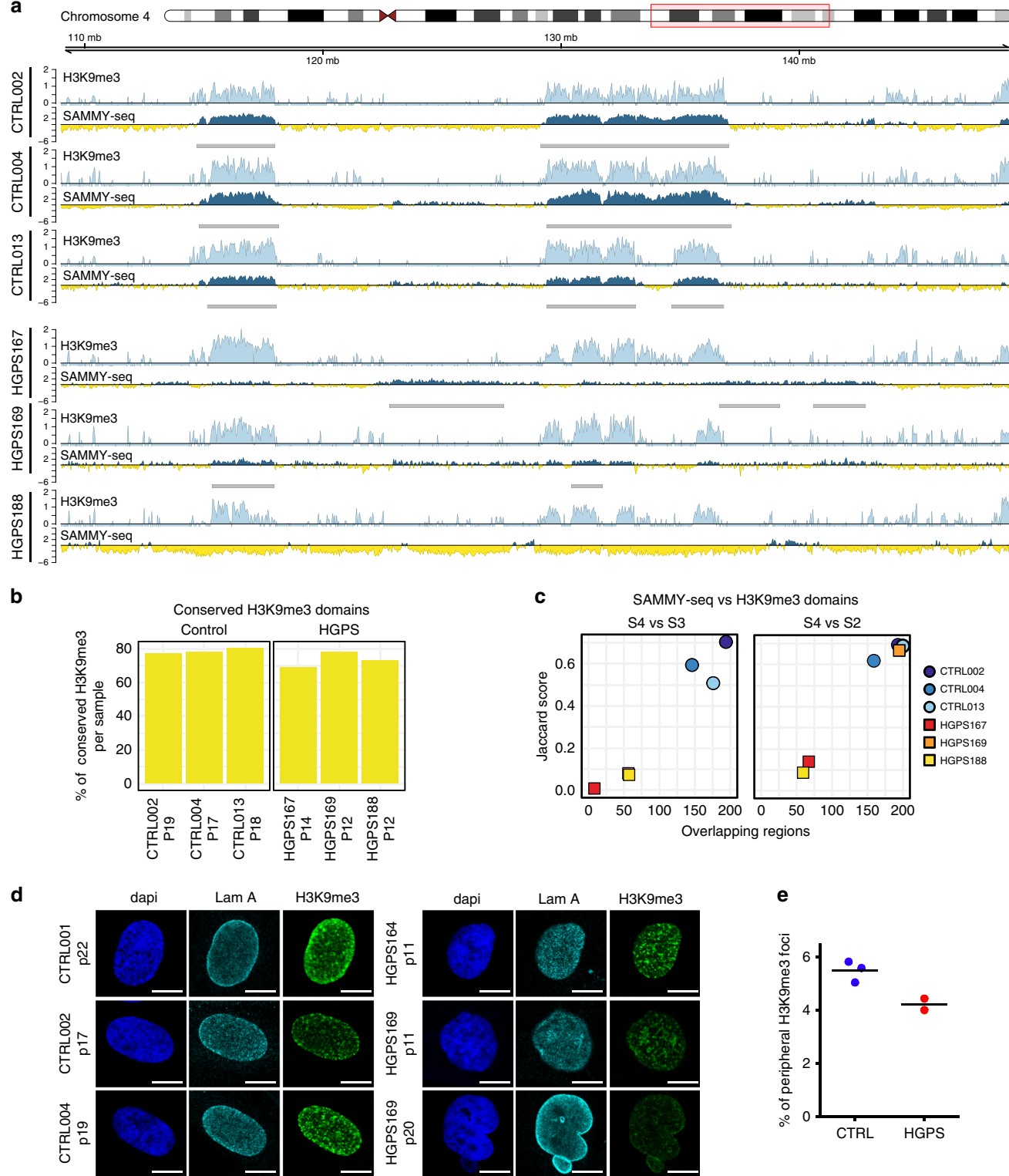

other and distant from the progeria samples, the latter also showing larger distances between them.

To further dissect the role of chromatin regulators in the disrupted expression patterns, we examined the overlap of SAMMY-seq domains with the chromatin states of healthy skin fibroblasts[49]. SAMMY-seq domains of control samples are mostly overlapping "Quiescent" or "Heterochromatin" states (Fig. 5d), whereas SAMMY-seq domains of HGPS cells comprise regions

normally transcribed or regulated by PcG proteins in healthy skin fibroblasts. To analyze the correlation between gene expression, chromatin states, and altered genome structure, we grouped SAMMY-seq domain genes on the basis of their chromatin state in healthy skin fibroblasts and examined their expression difference between control and HGPS cells (Fig. 6a and Supplementary Fig. 9c). This analysis revealed that PcG-regulated and bivalent genes are deregulated at the transcriptional

**Fig. 4 Early chromatin structure disruption in progeria is not accompanied by alterations in H3K9me3. a** Genomic tracks for paired H3K9me3 ChIP-seq and SAMMY-seq in three control and three HGPS samples for a representative region (40 Mb region in chr4:109000000-149000000). Enrichment signals (computed using SPP) for each sample are shown (H3K9me3 ChIP-seq – light blue; blue or yellow-colored track for SAMMY-seq S4 vs S2 enrichment or depletion, respectively). Gray boxes under each SAMMY-seq track show the SAMMY-seq domains. Number of passages in each sample for SAMMY-seq and ChIP-seq: CTRL002, P20 and P19; CTRL004, P20 and P17; CTRL013, P19 and P18; HGPS167, P13 and P14; HGPS169, P12 and P12; HGPS188, P12 and P12. For ChIP-seq data, the (ChIP – input) reads distribution was computed with SPP package and we are limiting the y axis range to zero as a minimum value. **b** Percentage of H3K9me3 domains conserved across all three control or all three progeria samples at indicated passages (p12-19) (computed as a percent over per sample total H3K9me3 domains size). **c** Overlap of H3K9me3 enriched domains and SAMMY-seq domains (S4 vs S3, S4 vs S2) for control or HGPS samples (JI on y axis, number of overlapping regions on x axis). All overlaps are significant in controls (for both S4 vs S3 and S4 vs S2 SAMMY-seq domains) (upper-tail Fisher test p-value < 0.01 for all overlaps) and in HGPS169 (only for S4 vs S2 SAMMY-seq domains). **d**, Representative images of H3K9me3/Lamin A immunofluorescence on 3 independent control replicates and two independent HGPS replicates at indicated passages (p11-22). Scale bars: 10 μm. **e** Percentage of H3K9me3 foci localized within 1 μm from the nuclear periphery in distinct cell populations (see "Methods" section). Each dot shows an independent biological replicate, the horizontal black line is the median. Number of analyzed cells/total bodies/number of bodies included in 1 μm from nuclear periphery, respectively, are reported below: CTRL001p22: 242/8598/434; CTRL002p17: 649/38160/2133; CTRL004p19: 518/34434/2006; HGPS164p11: 550/32885/1317; HGPS169p11: 65/2099/93.

level in progeria with respect to controls. To evaluate whether bivalent genes are especially affected in progeria, we performed a Fisher test: we found 76 bivalent out of 257 differentially expressed genes in HGPS (Fisher test p-value < 2.2e−16); or 39 bivalent out of 144 upregulated genes in HGPS (Fisher test p-value 4.32e−12). We thus analyzed the H3K27me3 profile of the subset of bivalent genes upregulated in HGPS, where we observed a clear drop in the H3K27me3 signal (Fig. 6b). These findings overall suggest that early chromatin remodeling has an impact on a subset of PcG targets, i.e., the bivalent genes, more susceptible to variations of PcG occupancy[45,61,62], as observed also at the level of H3K27me3 distribution by ChIP-seq (Fig. 6c).

To verify the presence of PcG-progerin crosstalk, we used the proximity ligation assay (PLA), which detects interactions between proteins in close proximity (<30 nm)[63]. We found that Lamin A/C interacts with PcG proteins in control or HGPS fibroblasts (Fig. 6d, e), and that progerin interacts with both Ezh2 and Bmi1 in late-passage HGPS (Fig. 6d–e). To further investigate the aberrant PcG-progerin interaction in HGPS progression, we tested PcG protein compartmentalization in both early and late-passage HGPS (Fig. 6f, g). When comparing controls and early-passage HGPS fibroblasts, we did not observe major differences in PcG nuclear distribution across chromatin fractions. On the other hand, PcG proteins were predominantly localized in the S4 fraction at late-passage HGPS. Altogether, our results indicate that an increased amount of progerin may alter PcG protein's nuclear compartmentalization and function.

## Discussion

Chromatin within the cell nucleus has a complex structure, which is fundamental for gene expression regulation[64]. In particular, heterochromatin is highly compacted and shows specific nuclear compartmentalization. LADs, along with pericentromeric and telomeric regions, are crucial for heterochromatin organization and for preserving chromosome structure[3,4,10]. Alterations in heterochromatin are associated with developmental defects and cancer, while its proper conformation is a hallmark of healthy cells[65,66]. As such, reliable methods to characterize heterochromatin and its alterations are essential for the biomedical scientific community.

Here we present a technology called SAMMY-seq, for genome-wide characterization of lamina-associated heterochromatic regions (Fig. 1). The protocol is based on the sequential extraction of multiple chromatin fractions, corresponding to increasingly compacted and less accessible chromatin regions, which are then mapped by high-throughput sequencing. All of the three SAMMY fractions are sequenced as they convey complementary information about chromatin structure, which may have fundamental

differences across distinct cell types[67]. SAMMY-seq can reproducibly and reliably identify lamina-associated heterochromatic regions as the less accessible portions of chromatin in control wild-type fibroblasts (Fig. 2). Moreover, SAMMY-seq represents a significant improvement in the field of chromatin characterization as it provides information complementary and not overlapping with other high-throughput sequencing-based methods commonly used to study chromatin structure and function (Supplementary Tables 1 and 5). Of note, our protocol overcomes several major limitations of other methods for mapping lamina-associated heterochromatic regions. First of all, the procedure can be applied to primary cells, as it does not require exogenous gene expression as in DamID-seq. Then, SAMMY-seq does not involve chemical modifications of chromatin, which might cause artifacts and biases in sequencing[68,69]. Additionally, it does not rely on antibodies for enriching specific chromatin fractions, thus avoiding issues related to antibody specificity, production, lot-variability, and cross-reactivity. This is particularly important when studying chromatin changes in cells where protein levels of chromatin-associated factors could be altered, thus allowing more flexibility in terms of experimental design compared to antibody-based techniques. Finally, SAMMY-seq is robust, as it yields reproducible results even at lower sequencing depth (Supplementary Fig. 1g) or with a small number of starting cells (Supplementary Fig. 3) and requires only about 3 h of bench work, excluding DNA extraction and library preparation. Given its characteristics, SAMMY-seq is a versatile technique that can be applied to different experimental systems, including primary cells and rare cell populations.

In early-passage HGPS fibroblasts, SAMMY-seq allowed us to detect changes in chromatin distribution across nuclear compartments characterized by distinct accessibility (Figs. 3 and 6h), despite the recent failure to do so by Hi-C[30]. Indeed, Hi-C is designed to measure pairwise interactions between genomic loci, but it is not able to distinguish the intranuclear compartments where such interactions occur[70].

Remarkably, changes in chromatin accessibility observed in early-passage HGPS are not accompanied by H3K9me3 alterations or transcriptional deregulation in the same genomic regions (Fig. 4 and Supplementary Figs. 6 and 7). Previous studies revealed that chromatin tethering at nuclear lamina and histone marks deposition are independent processes[71,72] and that nuclear relocation is not sufficient to trigger a transcriptional switch[72–74]. Overall, our findings support this scenario also in the HGPS model and further indicate that chromatin remodeling detected by SAMMY-seq precedes H3K9me3 decrease observed at later passages[54] (Fig. 4d).

On the other hand, we found that chromatin accessibility changes in HGPS functionally affect PcG-regulated genes, by

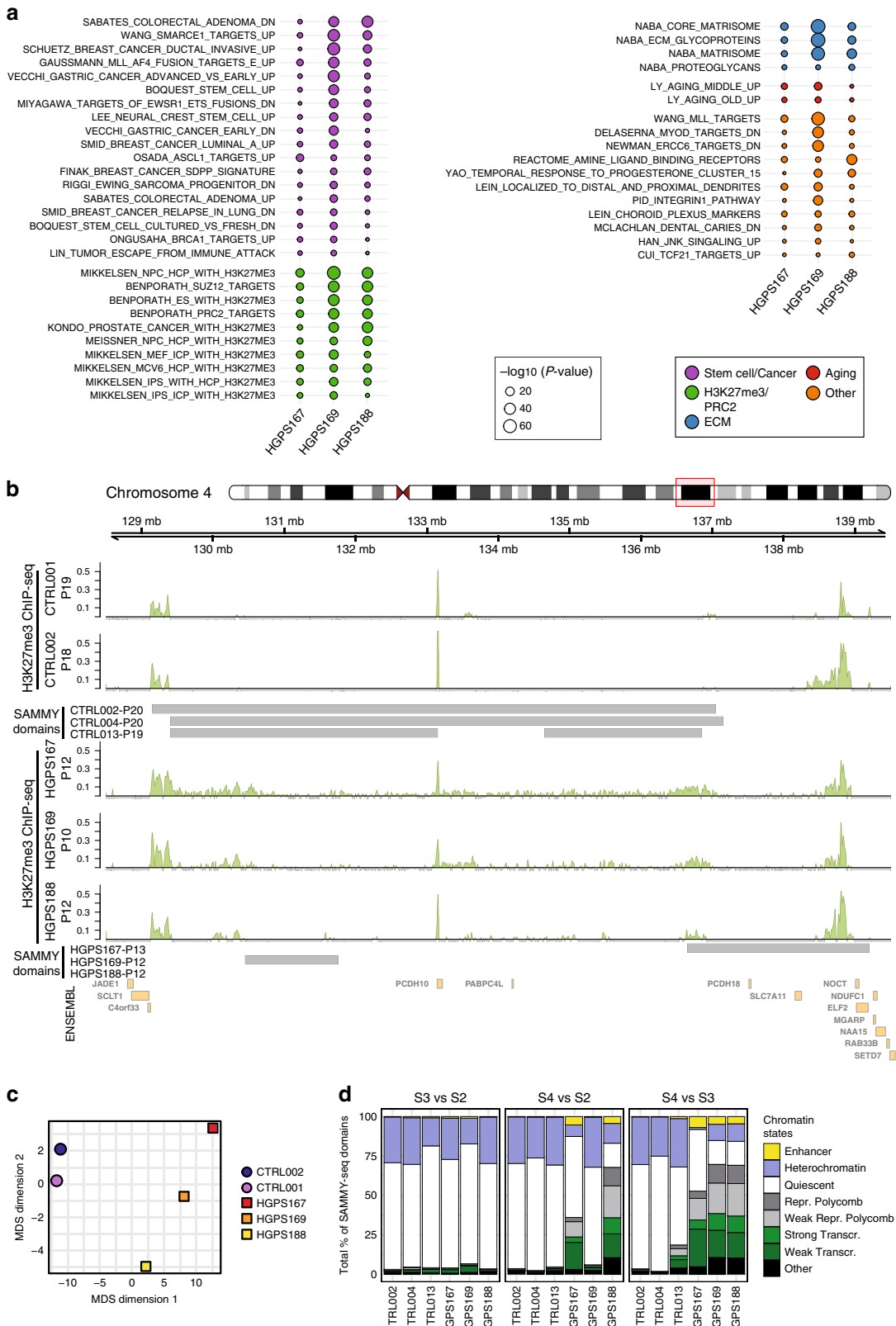

interfering with H3K27me3 genomic distribution (Fig. 5). The crosstalk between Lamin A and Polycomb, first reported by ourselves[19], is now supported by several independent studies[17,18,21–23]. The alteration of Polycomb regulation may be expected to first affect the so-called "bivalent" genes, which are already in an intermediate state, mostly silenced but primed for activation by the presence of both repressive (H3K27me3) and activating (H3K4me3) chromatin marks[45,61,62]. In fact, the bivalent genes, involved in stemness maintenance and cell identity specification, are mostly affected in early-passage HGPS fibroblasts (Fig. 6).

**Fig. 5 HGPS fibroblasts show an alteration of PRC2 distribution. a** MSigDB pathways consistently deregulated (FDR < 0.05) in all three progeria samples, based on comparison of individual HGPS samples to the group of control samples. A two-sided Wald test was used to compute p-values for the differential expression at the transcript level, then uncorrected p-values were aggregated at pathway level using the Lancaster method, and Benjamini–Hochberg FDR correction applied on the aggregated p-values (see "Methods" section). Pathways are ranked based on the sum of their −log10 transformed p-values across all three progeria samples. Dot colors highlight different categories of pathways as per graphical legend: related to stem cells or cancer, H3K27me3/PRC2 regulation, extracellular matrix (ECM), aging, or other categories. Dot size is proportional to the −log10 transformed p-value. **b** Genomic track for H3K27me3 ChIP-seq signal in control and HGPS samples at indicated passages (p10-19) for a representative region (11 Mb region in chr4:128500000 to 139500000). Enrichment signals (computed using SPP) for each sample are shown. SAMMY-seq domains (S4 vs S2 comparison) are reported as gray rectangles for three controls and three HGPS samples, as indicated in the labels. For ChIP-seq data, the (ChIP − input) reads distribution was computed with SPP package and we are limiting the y axis range to zero as a minimum value. **c** Multi-dimensional scaling of H3K27me3 ChIP-seq experiments using read counts in 10k genomic regions across all autosomes. The 2D distance of dots is representative of the pairwise distance of samples using the 10k genomic region read counts. **d** Overlap of SAMMY-seq domains from each sample to Roadmap Epigenomics chromatin states for fibroblasts (E055). The overlaps with each chromatin state are reported as a percentage over the total of SAMMY-seq domains (S3 vs S2, S4 vs S2 or S4 vs S3) for each sample.

---

Finally, it is worth remarking that this model would provide a mechanism for the previously postulated idea that the reduced potential for tissue homeostasis, commonly associated with progeria, is due to a misregulation of stem cell compartments[60].

## Methods

**Cell cultures**. Primary fibroblast cell lines were cultured in DMEM High glucose with glutamax supplemented with 15% FBS and 1% Pen/Strep. HGADFN164 (HGPS164)-4 years old, HGADFN167 (HGPS167)-8 years old, HGADFN169 (HGPS169)-8 years old, HGADFN188 (HGPS188)-2 years old, HGADFN271 (HGPS271) -1-year-old human dermal fibroblasts derived from HGPS patients were provided by the Progeria Research Foundation (PRF). AG08498 (CTRL001) -1-year-old and AG07095 (CTRL002)-2-years-old human dermal fibroblasts were obtained from the Coriell Institute. Preputial fibroblast strain #2294 (CTRL004)-4 years old was a generous gift from the Laboratory of Molecular and Cell Biology, Istituto Dermopatico dell'Immacolata (IDI)-IRCCS, Rome, Italy, while control dermal fibroblast CTRL013-13 years old was kindly provided by the Italian Laminopathies Network.

**Histochemistry, immunofluorescence assay, and PLA analysis**. Beta-galactosidase assay: Cells were fixed at room temperature for 10 min in paraformaldehyde at 4%, washed twice with 1×PBS, and incubated with fresh staining solution at 37 °C for 16 h in dark room. Then cells were washed twice with 1×PBS, overlaid with 70% glycerol/PBS, and photographed.

BrdU (5-Bromo-2¢-deoxy-uridine, Sigma B9285) labeling: Cells were grown for 8 h in the presence of 50 μM of BrdU, and then fixed in 4% PFA. After Triton X-100 treatment, cells were incubated for 2 min at RT in 0.07 N NaOH, briefly rinsed twice in PBS, and blocked in PBS/1% BSA. Reaction with BrdU antibody (1:10, Becton Dickinson 347580) was performed at room temperature for 1 h in a PBS solution containing BSA 1%.

Immunofluorescence assay: Coverslips were fixed with paraformaldehyde at 4% in PBS for 10 min. Then, cells were permeabilized with 0.5% Triton X-100 in PBS and non-specific signals were blocked with 1% BSA in PBS for 30 min at room temperature. The following antibodies were used: Bmi1 (Millipore 05-637, mouse) diluted 1:100; Lamin A/C (Santa Cruz sc-6215, goat) diluted 1:200; Ezh2 (Cell signaling AC22 3147 S, mouse) diluted 1:100; H3K9me3 (Abcam ab8898, rabbit) diluted 1:500; H3K27me3 (Millipore 07-449, rabbit) diluted 1:100. Incubation was performed at 12–16 h at 4 °C or at room temperature for 2 h for Lamin A/C. Primary antibodies were diluted in a PBS solution containing BSA 1%. Cells were stained with appropriate secondary antibodies, diluted 1:200 for 1 h at room temperature. Washes were done in PBT. As secondary antibodies, we used Alexa Fluor 488 Donkey anti-mouse IgG (715-545-150), Alexa Fluor 594 Donkey anti-goat IgG (705-585-003), Alexa Fluor 488 Donkey anti-rabbit IgG (711-545-152) from Jackson ImmunoResearch Laboratories, and Alexa Fluor 647 Chicken anti-goat IgG (Invitrogen, A21469). Finally, DNA was counterstained with DAPI, and glasses were mounted in Vectashield Antifade (Vector Laboratories) or ProLong Gold Antifade Reagent (Invitrogen). For PLA experiments, coverslips were fixed first with paraformaldehyde at 4% in PBS for 10 min. Then, cells were permeabilized with 0.5% Triton X-100 in PBS and blocked with 1% BSA in PBS for 1 h at room temperature. Incubation with progerin (Alexis human mAb, 13A4, ALX-804-662-R200, mouse) diluted 1:20 or Lamin A/C (Santa Cruz sc-6215, goat) diluted 1:200 was performed for 12–16 h at 4 °C. After PBT washes, cells were incubated with Ezh2 (Cell signaling 4905S, rabbit 1:100) or Bmi1 (Abcam ab85688 rabbit 1:100) for 12–16 h at 4 °C. Primary antibodies were diluted in a PBS solution containing BSA 1%. Finally, detection of protein interactions was performed using the Duolink system (Sigma) following the manufacturer's instructions.

**Image processing and analysis**. Fluorescent images were taken with a Nikon ECLIPSE 90i microscope (1006 objective), equipped with a digital camera (Nikon

Coolpix 990) and NIS Element Digital software (Supplementary Fig. 4b and data collected for Supplementary Fig. 4c) or with confocal Leica SP5 supported by LAS-AF software (version 2.6.0) (Figs. 4d, 6d, e and Supplementary Figs. 4d, 8e, g). Confocal image size was 1024×1024 with the following laser wavelengths: 405-Diode; Argon; HeNe-633. Argon laser emission power was set to a range of 30–50% of total power amplitude. For acquisition settings, excitation power was set at 30% of Argon; 52% of 405-Diode, and 25% of HeNe-633. PLA blob quantification was performed using Cell Profiler 2.0. Nuclei were detected using the Otsu method with a global two-class threshold strategy, and foci were detected using the Otsu method with a per-object three-class threshold strategy[75]. In order to automatically detect and quantify PcG bodies in fluorescence cell images, we improved a method derived from a previous work[76].

The algorithm has been implemented in MATLAB[77] following the pseudocode reported in Supplementary Methods.

The algorithm computes the number of PcG bodies, the area of any PcG body, and the minimum Euclidean distance of any PcG body from the nuclear periphery. In order to compare these minimum distances among nuclei of different sizes, we compute for each PcG body a measure of its closeness to the nuclear periphery (proximity). This measure is size-independent, being computed by dividing the minimum distance by the distance of the nuclear centroid from the point on the nuclear periphery closest to the PcG body.

**Protein extraction and western blot analyses**. Total proteins were prepared by resuspending $1 \times 10^6$ cells in extraction buffer (50 mM Tris-HCl pH 7.6; 0.15 M NaCl; 5 mM EDTA; 16 Protease Inhibitors; 1% Triton X-100). One 40 s pulse of sonication (UP100H manual sonicator, Hielscher) at 40% amplitude was performed to allow dissociation of protein from chromatin and solubilization. Extracts were analyzed by SDS-PAGE using an 8% gel (37.5:1 Acryl/Bis Acrylamide). The following primary antibodies were used: Beta-Actin (Santa-Cruz sc1616, rabbit 1:4000), H3 total (Abcam ab1791, rabbit 1:6000), Lamin A/C (Santa Cruz sc-6215, goat 1:4000), Lamin B (Santa Cruz sc6216, goat 1:2000), progerin (13A4 mouse, Abcam 66587, mouse 1:1000), Ezh2 (AC22 Cell Signaling 3147S, mouse 1:1000), Bmi1 (D42B3 Cell signaling, rabbit 1:1000), H3K9me3 (Abcam ab8898, rabbit 1:1000), H3K27me3 (Millipore 07-449 rabbit 1:1000). HRP-conjugated secondary antibodies were revealed with the ECL chemiluminescence kit (ThermoFisher Scientific). The following secondary antibodies were used: Anti-Mouse IgG-Peroxidase (Sigma, A9044), Anti-Rabbit IgG-Peroxidase (Sigma, A9169), Anti-Goat IgG-Peroxidase (Sigma, A5420).

**Chromatin fractionation**. Chromatin fractionation was carried out as described in ref. [47] with minor adaptions. Briefly, 4 million fibroblasts were washed in PBS 1×, and resuspended in cytoskeleton buffer (CSK: 10 mM PIPES pH 6.8; 100 mM NaCl; 1 mM EGTA; 300 mM Sucrose; 3 mM MgCl₂; 1× protease Inhibitors by Roche Diagnostics; 1 mM PMSF) supplemented with 1 mM DTT and 0.5% Triton X-100. After 5 min at 4 °C the cytoskeletal structure was separated from soluble proteins by centrifugation at 900×g for 3 min, and the supernatant was labeled as S1 fraction. The pellets were washed with additional volume of cytoskeleton buffer. Chromatin was solubilized by DNA digestion with 10U of RNase–free DNase (Turbo DNAse; Invitrogen AM2238) in CSK buffer for 60 min at 37 °C. To stop digestion, ammonium sulfate was added in CSK buffer to a final concentration of 250 mM and, after 5 min at RT samples were pelleted at 2350×g for 3 min at 4 °C and the supernatant was labeled as S2 fraction. After a wash in CSK buffer, the pellet was further extracted with 2 M NaCl in CSK buffer for 5 min at 4 °C, centrifuged at 2350×g 3 min at 4 °C and the supernatant was labeled as S3 fraction. This treatment removed the majority of histones from chromatin. After 2 washing in NaCl 2 M CSK, the pellets were solubilized in 8 M urea buffer to remove any remaining protein component by applying highly denaturing conditions. This remaining fraction was labeled as S4. For the scaled-down experiment, samples of 25,000, 50,000, or 10,000 cells were treated analogously, except for a reduction of

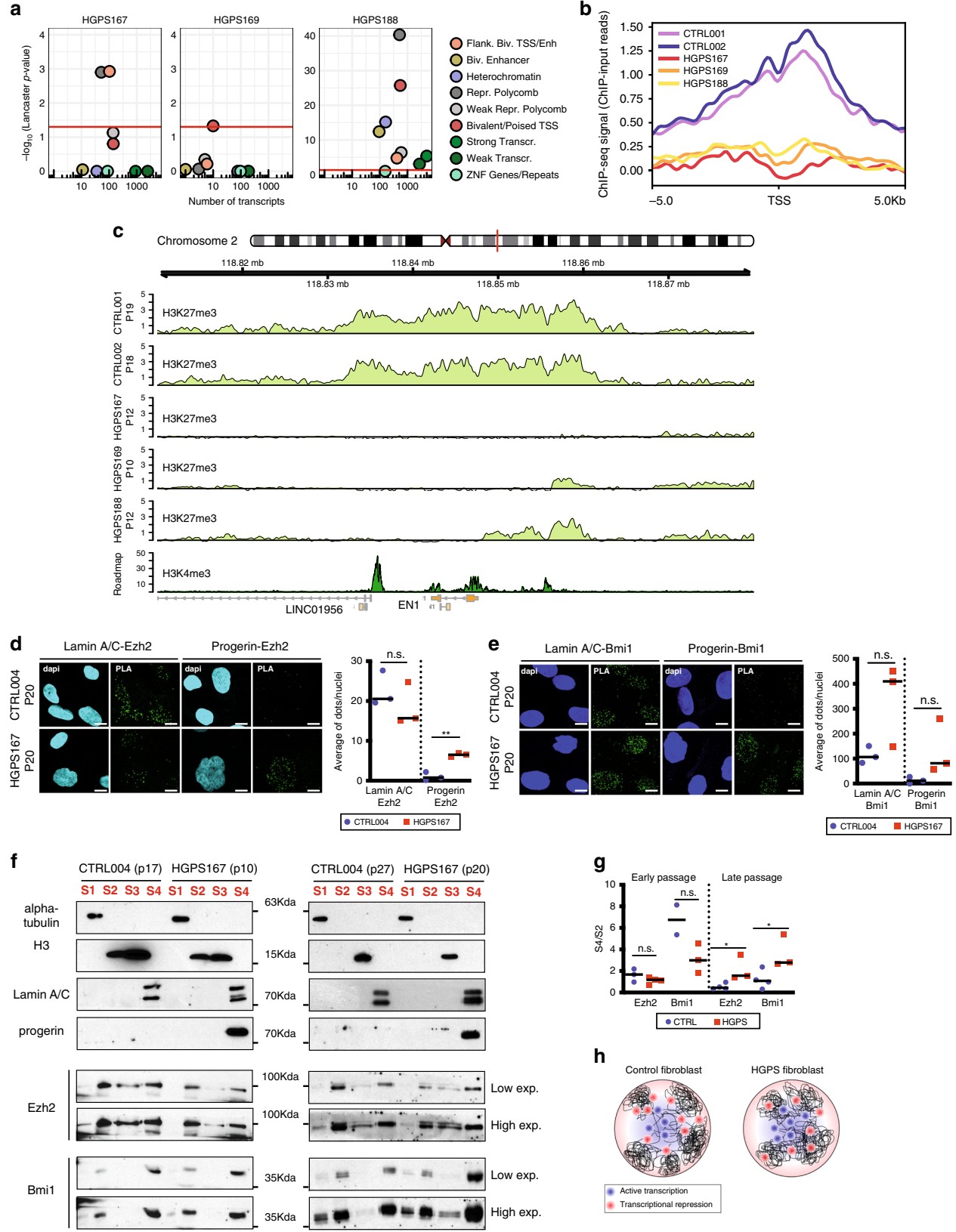

buffers volumes to half of those used for 10 million cells and a decrease of DNase to 8U.

For protein analysis, 70% of the volume of each fraction was and analyzed by SDS-PAGE and immunoblotting. Anti-tubulin alpha (Sigma T5168, mouse 1:10,000), H3 (Abcam ab1791, rabbit 1:4000), Beta-Actin (Santa-Cruz sc1616, rabbit 1:4000), Lamin A/C (Santa Cruz sc-6215, goat 1:4000), Lamin B (Santa Cruz sc6216, goat 1:2000), progerin (13A4 mouse, Abcam 66587, mouse 1:1000), Ezh2

(AC22 Cell Signaling 3147S, mouse 1:1000), Bmi1 (D42B3 Cell signaling, rabbit 1:1000) were used as primary antibodies. HRP-conjugated secondary antibodies were revealed with the ECL chemiluminescence kit (Thermo Fisher Scientific).

**DNA sonication and sequencing for chromatin fractionation.** For DNA analysis, the remaining 30% of the volume of each fraction was diluted to 300 μl with TE,

**Fig. 6 Chromatin structural changes in HGPS affect PRC2-regulated bivalent genes. a** Transcript-level differential expression (upregulation) was assessed by comparing individual HGPS samples against the group of controls. A two-sided Wald test was used to compute $p$-values for the differential expression at the transcript level, then uncorrected $p$-values were aggregated based on their chromatin state using the Lancaster method, considering only genes in SAMMY-seq domains (S4 vs S2). The plot reports the number of transcripts ($x$ axis) and significance ($y$ axis) for each chromatin state (color legend) as defined by ref. [49]. The horizontal red lines mark the 0.05 $p$-value threshold. **b** Average H3K27me3 ChIP-seq signal ($y$ axis) around the Transcriptional Start Site (TSS) of 39 bivalent genes upregulated in HGPS. The $x$ axis reports relative genomic position around the TSS (±5 kb). **c** Representative genomic tracks (chr2: 118810000-118880000) of H3K27me3 ChIP-seq in control and HGPS samples at indicated passages (p10-19) around the bivalent gene *EN1* upregulated in HGPS samples. H3K4me3 ChIP-seq from Roadmap Epigenomics normal skin fibroblasts is reported as well. The (ChIP – input) reads distribution was computed with SPP package and we are limiting the $y$ axis range to zero as a minimum value. **d, e** Representative images of Proximity Ligation Assay (PLA) experiments in CTRL004 and HGPS167 at late passages (p20). Each fluorescent dot represents the co-localization of Lamin A/C or Progerin and Ezh2 (**d**) or Bmi1 (**e**). Nuclei were stained with dapi. Scale bars: 20 μm. The PLA acquisition parameters were the same in control and HGPS cells for each interactor pairs. All data were generated from an average of three independent experiments, the horizontal black line is the median. Each dot shows an independent biological replicate. Exact $p$-value: 0.0011. **f** Representative western blot on chromatin fractionation experiments of CTRL004 and HGPS167 at indicated early (left, p10-17) and late (right, p20-27) passages. Equal amounts of each fraction were hybridized with the indicated antibodies. Alpha-tubulin, histone H3, and Lamin A/C were used as loading controls respectively for S1, S2 and S3, S4. **g**, The graph shows quantifications of Ezh2 and Bmi1 bands of at least two independent biological replicates reporting S4 fraction normalized on S2, the horizontal black line is the median. Each dot shows an independent biological replicate. Exact $p$-values Ezh2: 0.041; Bmi1: 0.044 **h**, Cartoon illustrating the proposed model: that expression of progerin in early-passage HGPS fibroblasts interferes with LADs positioning near the nuclear envelope and is accompanied by altered transcriptional profiles at PcG-regulated heterochromatin. Comparisons were done using a two-tail $t$-test in **d, e, g**. Statistically significant differences are marked *$p < 0.05$; **$p < 0.01$.

---

incubated with RNAse A (Roche) (90 min at 37 °C) and Proteinase K (Sigma) (150 min at 55 °C), DNA was extracted by standard phenol/chloroform extraction, precipitated and resuspended in 25 μl Milli-Q H$_2$O. After Nanodrop (260/280 = 1.7–1.9; 260/230 ≥ 2) and Qubit HS DNA quantification, we added H$_2$O to a final volume of 105 μl. Then samples were transferred to 96 well plates and sonicated 4 times with Bioruptor sonicator (10 min 30 s ON - 30 s OFF, High Power). The DNA profiles were finally checked by capillary electrophoresis (Agilent 2100 Bioanalyzer with 2100 Expert Software). Finally, DNA libraries were prepared by using NuGEN Ovation Ultralow Library Prep System kit and then sequenced using an Illumina HiSeq 2500 instrument according to manufacturer's instructions (Illumina). High passage samples were sonicated using Bioruptor or Covaris sonicator to reach a fragment dimension range of 100–500 bps. Samples were then quantified using Qubit HS DNA kit (Life Technologies) and library prepared from each of them using NEBNext Ultra II DNA Library Prep Kit for Illumina (NEB, E7645L) with Unique Dual Index NEBNext Multiplex Oligos for Illumina (NEB, E6440S). The library was quantified using Qubit HS DNA kit (Life Technologies) and profiles were checked by capillary electrophoresis on an Agilent 2100 Bioanalyzer. Finally, DNA libraries were sequenced using an Illumina NovaSeq 6000 instrument at the IEO Genomic Unit in Milan. In the scaled-down experiment, due to the low amount of starting material, the entire volume of each fraction was diluted to 200 μl with TE and used for DNA extraction. After extraction, S2 fractions were further purified using a PCR DNA Purification Kit (Qiagen). DNA from each fraction was resuspended in nuclease-free water, transferred to microTUBE-15 AFA Beads Screw-Cap (Covaris), and sonicated in a Covaris M220 focused-ultrasonicator (water bath 20 °C, peak power 30.0, duty factor 20.0, cycles/burst 50, duration: 125 s for S2 fraction, 150 s for S3 and S4 fractions). S2 fractions of 10k and 50k samples were not sonicated because their fragments were already in the range for library generation. As for high passage samples, libraries were prepared with NEBNext Ultra II DNA Library Prep Kit for Illumina and Unique Dual Index NEBNext Multiplex Oligos for Illumina and they were sequenced using an Illumina NovaSeq 6000 instrument at the IEO Genomic Unit in Milan.

**Chromatin immunoprecipitation and sequencing.** Cells were cross-linked with 1% HCHO for 12 min at room temperature, lysed and chromatin sheared. IP was performed overnight on a wheel at 4 °C with 2.4 μg of H3K9me3 antibody (ab8898, Abcam) or 2μg of H3K27me3 (07-449, Millipore). The following day, antibody-chromatin immunocomplexes were loaded onto Dynabeads Protein G (Invitrogen 10004D). The bound complexes were washed once in Low Salt Solution (0,1% SDS, 2 mM EDTA, 1% Triton X-100, 20 mM Tris pH 8, 150 mM NaCl), once in High Salt Solution (0,1% SDS, 2 mM EDTA, 1% Triton X-100, 20 mM Tris pH 8, 500 mM NaCl), once again in Low Salt Solution and once in Tris/EDTA 50 mM NaCl. Crosslinking was reversed at 65 °C overnight in Elution Buffer (50 mM Tris pH 8, 20 mM EDTA, 1%SDS), DNA was purified by standard phenol/chloroform extraction, precipitated and resuspended in 30 μl of 10 mM Tris pH 8. ChIP efficiency was tested by qPCR reactions, performed in triplicate using SYBR select master mix (Invitrogen, 4472908) on a StepOnePlus™ Real-Time PCR System (Applied Biosystems). Primer sequences are reported in Supplementary Table 7. Relative enrichment was calculated as IP/Input ratio. For H3K9me3, ChIP-seq libraries for sequencing were created using the automation instrument Biomek FX (Beckman Coulter), while for H3K27me3 ChIP-seq they were created using NEBNext Ultra II DNA Library Prep Kit for Illumina (NEB, E7645L) with NEB-Next Multiplex Oligos for Illumina (NEB). Libraries were then qualitatively and

quantitatively checked using dsDNA HS Assay kit (Invitrogen, Q32854) on a Qubit 2.0 fluorometer and High Sensitivity DNA Kit (Agilent Technologies, 5067–4627) on an Agilent Bioanalyzer 2100. Libraries with distinct adapter indexes were multiplexed and, after cluster generation on FlowCell, were sequenced for 50 bp in the single read mode on a HiSeq 2000 sequencer at the IEO Genomic Unit in Milan.

**RNA sequencing.** For high-throughput sequencing, cDNA libraries were prepared from total RNA, extracted with Trizol, by using Illumina TruSeq Stranded Total RNA Kit with Ribo-Zero GOLD. cDNA fragments of ~300 bp were purified from each library and were sequenced for 125 bp, using an Illumina HiSeq 2500 instrument according to manufacturer's instructions (Illumina).

**SAMMY-seq sequencing read analysis.** Sequencing reads were aligned to the hg38-noalt reference human genome available in the bcbio-nextgen pipeline, using bwa aln[78] (version 0.7.12) with options -n 2 -k 2 and saved the results in sam format with bwa samse. The sam files were converted to bam and name sorted with samtools[79] (version 1.3.1). We marked PCR duplicates using the biobambam2 toolset[80] (version 2.0.54). We discarded reads mapping to non-autosomal chromosomes, PCR duplicates, qcfail, unmapped, or mapping quality 0 reads with samtools. We converted the bam files to bedgraph using bedtools[81] (version 2.25.0) and bedgraph to bigWig using UCSC's bedgraphToBigWig tool (version 4) for reads distribution visualization. Read coverage was normalized by the total sequencing library size, before converting bedgraph files to bigWig. Sample quality was assessed using fastqc (version 0.11.5) (http://www.bioinformatics.babraham.ac.uk/projects/fastqc). For the down-sampling analysis (Supplementary Fig. 1g), we sampled the raw fastq files to 50 and 25% using the seqtk sample command (https://github.com/lh3/seqtk) and ran the same pipeline, as above.

For the SAMMY-seq scale-down (Supplementary Fig. 3) and late passage (Supplementary Fig. 5) data sets analysis, we used the same pipeline as above with newer software sub-versions for bwa (version 0.7.17), samtools (version 1.9) and bedtools (version 2.29.0).

**ChIP-seq sequencing read analysis.** Sequencing reads were aligned, filtered, converted, and quality checked using the same tools as for the SAMMY-seq reads. Before sequence alignment, we did an additional trimming step, using Trimmomatic[82] (version 0.32) in single-end mode for the H3K9me3 data. We used the Enriched Domain Detector (EDD, version 1.1.15)[48] tool to call H3K9me3 peaks, with the following options:–fdr 0.1 –gap-penalty 10 –bin-size 100 –write-log-ratios –write-bin-scores and also excluding blacklisted genomic regions containing telomeric, centromeric, and certain heterochromatic regions[83]. We also changed the required_fraction_of_informative_bins parameter to 0.98.

We calculated genome-wide ChIP-seq signal for H3K9me3 and H3K27me3 data, using the SPP package (version 1.15.4)[84]. We imported bam files into the R (version 3.3.1) statistical environment, and selected informative reads with the get.binding.characteristics and select.iformative.tags functions. The remove.tag.anomalies function removed anomalous positions with an extremely high number of reads using the remove.tag.anomalies function, and calculated the differential signal, smoothed by a Gaussian kernel, using the get.smoothed.tag.density function with the default bandwidth and tag.shift parameters. In the case of H3K27me3 data, we also set the scale.by.data.set.size = TRUE parameter for the get.smoothed.tag.density function. The output of the get.smoothed.tag.density function is an input subtracted (ChIP – input)

Gaussian smoothed sequencing reads distribution, and we refer to this as "ChIP-seq signal" in the genomic tracks and metaprofiles.

We defined H3K27me3 containing gene TSS regions as follows. We called ChIP-seq peaks with the get.broad.enrichment.clusters function of SPP on the preprocessed data with window.size = 2000 and z.thr = 3, tag.shift = ((cross correlation peak position)/2) parameters. We defined a set of peak regions present in all control samples using the bedtools intersect command and selected those protein coding genes, that had a common H3K27me3 peak region ±500 nt around the TSS. We defined a list of 4101 genes this way, which were used in downstream analyses.

MDS analysis was done using the plotMDS function of edgeR (version 3.24.3)[85] after importing the 10k genomic region read counts for all samples into the R (version 3.5.1) statistical environment. We dropped all genomic regions where the log count-per-million (cpm) value did not reach at least 1 in at least 2 samples. We calculated the normalization factor of samples using the calcNormFactors and estimated dispersion using the estimateDisp function of edgeR. We used the top = 1000 and gene.selection = "common" parameters for the plotMDS function. The gene.selection parameter here does not refer to genes, but the imported genomic regions.

We used deepTools (version 3.2.1)[86] to visualize the ChIP-seq signal around TSS regions. We used the following parameters for the computeMatrix reference-point subcommand: -b 5000 -a 5000 –binSize 100, to calculate a matrix of SPP signal values per bin and gene. For visualization, we used the plotProfile command.

**Literature data processing**. We collected publicly available data sets from the following sources: ATAC-seq from[87] (GEO: GSE80639), Lamin A/C ChIP-seq from[30,48] (GEO: GSE41757 and GSE54332), Lamin B1 ChIP-seq from[48,51,52] (GEO: GSE49341 and GSE63440). Sequencing reads were aligned, filtered, and converted using the same tools as for the SAMMY-seq reads. We calculated the genome-wide ChIP-seq signal for Lamin A/C and Lamin B1 data, using the SPP package (version 1.16.0)[84]. We imported bam files into the R (version 3.5.1) statistical environment, and selected informative reads with the get.binding.characteristics and select.informative.tags functions, removed anomalous positions with an extremely high number of reads using the remove.local.tag.anomalies function, and calculated the differential signal, smoothed by a Gaussian kernel, using the get.smoothed.tag.density function with the default bandwidth and tag.shift parameter. We used the filtered read count as the genome-wide signal for the ATAC-seq sample.

We downloaded from Roadmap Epigenomics[49] the aligned reads of the consolidated data sets for histone marks ChIP-seq (H3K9me3, H3K27me3, H3K4me1, H3K36me3, H3K27ac, H3K4me3) for the foreskin fibroblast sample (E055 consolidated data set in TagAlign format were retrieved from the Roadmap Epigenomics on line data repository at URL https://egg2.wustl.edu/roadmap/data/byFileType/alignments/consolidated). We converted the TagAlign files from hg19 to hg38 with the liftOver tool, then processed the reads to obtain the genome-wide ChIP-seq signal with SPP as described above for Lamin ChIP-seq samples, and we applied the get.smoothed.tag.density function with background.density.scaling = TRUE. We started from the aligned reads instead of the raw FASTQ reads so as to take advantage of the uniformly resampled data prepared by the Roadmap Epigenomics consortium, and to ensure maximum comparability with their originally published results, while at the same time processing the data with a pipeline comparable to what we used for Lamin ChIP-seq data, as well as ours ChIP-seq data sets. In addition, we downloaded the genome-wide signal coverage track for the DNAse-seq for the foreskin fibroblast sample E055. We converted the bigwig file to bedgraph with the bigWigToBedGraph tool, lifted over genomic coordinates from hg19 to hg38 with the liftOver tool, and converted back bedgraph to bigwig with the bedGraphToBigWig tool using the UCSC toolkit[88] (version 4).

We calculated genome-wide correlations between SAMMY-seq samples and all of the public data sets listed and processed above using StereoGene[53] (version 2.20). StereoGene uses kernel correlation to calculate genome-wide correlation for spatially related but not completely overlapping features irrespective of their discrete or continuous nature.

We called lamina-associated domains with the EDD tool (version 1.1.19). For the Lamin A/C data set, we used the following options: –gap-penalty 25 –bin-size 200 –write-log-ratios –write-bin-scores, and for the Lamin B1 data set, we used: –gap-penalty 5 –bin-size 100 –write-log-ratios –write-bin-scores. We also changed the required_fraction_of_informative_bins parameter to 0.98.

For Lamin A/C and Lamin B1, we defined a consensus set of regions for each of them (LADs) for specific downstream analyses with bedops[89] (version 2.4.37), by selecting those genomic regions, that were called LADs by the EDD tool in at least two different ChIP-seq experiments (for either Lamin A/C or Lamin B1 data sets). We used a total of 6 ChIP-seq experiments for Lamin A/C and 5 for Lamin B1. We used these consensus regions for downstream analysis in Fig. 2c and Supplementary Fig. 2b, c.

**SAMMY-seq domain analyses and visualization**. We performed relative comparisons of SAMMY-seq fractions within each sample using EDD (version 1.1.15), optimized to call very broad enrichment domains. EDD was originally designed for lamin ChIP-seq data, comparing IP and input samples. As SAMMY-seq data also shows broad enrichment regions, we used EDD to select significantly enriched

SAMMY-seq domains by comparing less accessible fractions to more accessible ones (S3 vs S2, S4 vs S3, and S4 vs S2 comparisons) in each sample, with the following options: –gap-penalty 25 –bin-size 50 –write-log-ratios –write-bin-scores and also excluding blacklisted genomic regions containing telomeric, centromeric, and certain heterochromatic regions[83]. We also changed the required_fraction_of_informative_bins parameter to 0.98. We used the same set of parameters for the downsampled data (Supplementary Fig. 1g), except changing –bin-size to 100 for 50% down-sampling or 200 for 25% down-sampling. To account for the lower sequencing depth, in the SAMMY-seq scale-down experiment (results reported in Supplementary Fig. 3), we run EDD (version 1.1.19) with –gap-penalty 25 and –bin-size 250 parameters, and in late-passage cells (results reported in Supplementary Fig. 5), we used –gap-penalty 45 and –bin-size 200.

Additionally, we calculated the genome-wide differential signal for all comparisons, using the SPP package (version 1.15.4)[84]. We imported bam files into the R (version 3.3.1) statistical environment, and selected informative reads with the get.binding.characteristics and select.informative.tags functions, removed anomalous positions with an extremely high number of reads using the remove.local.tag.anomalies function, and calculated the differential signal, smoothed by a Gaussian kernel, using the get.smoothed.tag.density function with the default bandwidth parameter and tag.shift = 0.

We used bedtools[81] (version 2.25.0) and the bedtools jaccard or bedtools fisher command to calculate Jaccard Index or Fisher test p-values for the various overlap analyses. We randomized SAMMY-seq domains and LADs for the SAMMY-seq – LAD overlap analysis using the bedtools shuffle with -noOverlapping option and also excluding blacklisted genomic regions containing telomeric, centromeric, and certain heterochromatic regions[83] with the -exclude option.

For the SAMMY-seq scale-down and late-passage data sets analysis, we used newer software sub-versions for R (version 3.5.1), the SPP package (version 1.16.0), and bedtools (version 2.29.0).

We used the Gviz (version 1.26.5)[90] and karyoploter (version 1.2.2)[91] Bioconductor packages to visualize SAMMY-seq read coverages, differential signals, and domains. We used ggplot2 (version 3.3.2)[92] for additional plotting, and the generalized additive model (GAM) method[93] from the mgcv package (version 1.8–12) for smoothing data while plotting SAMMY-seq border region signals.

**RNA-seq sequencing read analysis**. Transcript and gene-level quantification were done with Kallisto[94] (version 0.43.0) to estimate transcript-level read counts and TPM (Transcripts Per Million) values. The GENCODE v27 annotation was used to build the transcriptome index. Kallisto was run with the–bias option to perform sequence-based bias correction on each sample. We used sleuth[95] (version 0.29.0) to calculate gene or transcript-level differential expression comparing all three controls against all three HGPS samples, where the linear model used in sleuth included the sample type (control or HGPS), sex, and library id. Additionally, we analyzed transcript-level differential expression, comparing the three controls with a single progeria sample, where the linear model used in sleuth included the sample type (control or HGPS) and sex. For both cases, we used sleuth's Wald test to calculate p and q-values. After differential expression, we grouped transcripts according to MSigDB pathways and aggregated transcript p-values using the Lancaster aggregation method from the aggregation R package (version 1.0.1) motivated by a recently described analysis pipeline in ref. [96]. Lancaster p-values were corrected for multiple testing using the Benjamini–Hochberg method. Alternatively, we aggregated transcript p-values based on their Roadmap chromatin states and differential expression direction. We used two different criteria: for the Tx, TxWk, ReprPC, ReprPCWk, Het, and ZNF/Rpts Roadmap chromatin states, we filtered for transcripts overlapping at least 50% with the state. For the EnhBiv, TssBiv, and BivFlnk Roadmap chromatin state regions, we required that the 200 nt region around the TSS region overlaps at least 50% with the state, and these were considered bivalent genes for specific downstream analyses. We aggregated p-values of upregulated (sleuth b-value > 0) or downregulated (sleuth b-value < 0) transcripts separately, according to their chromatin state overlap and additionally filtering for being in a specific (S4 vs S3 or S4 vs S2) enrichment region.

**Reporting summary**. Further information on research design is available in the Nature Research Reporting Summary linked to this article.

## Data availability

The high-throughput sequencing data generated for this study are available in the GEO repository with accession number "GSE118633". Other previously published genomics data used in this article were released in public repositories by the original publication authors as indicated in the article or Methods details above. These include GEO data sets for ATAC-seq (GSE80639), Lamin A/C ChIP-seq (GSE41757 and GSE54332), Lamin B1 ChIP-seq (GSE49341 and GSE63440), in addition to Roadmap Epigenomics consolidated ChIP-seq data sets for E055 (human foreskin fibroblasts) for histone marks (H3K9me3, H3K27me3, H3K4me1, H3K36me3, H3K27ac, and H3K4me3) retrieved from the Roadmap Epigenomics on line repository at URL [https://egg2.wustl.edu/roadmap/data/byFileType/alignments/consolidated]. All other relevant data supporting the key findings of this study are available within the article and its Supplementary Information files or from the corresponding authors upon reasonable request. A reporting summary for this Article is available as a Supplementary Information file. Source data are provided with this paper.

## Code availability

The Matlab code for the analysis of images is available in a Github repository at https://doi.org/10.5281/zenodo.4016157.

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

## Acknowledgements

We thank Giovanna Lattanzi, Sammy Basso, the Italian network of Laminopathies, and members of the laboratory for stimulating discussions and constructive criticisms. We thank Beatrice Bodega, Marina Lusic, Maria Vivo, Daniela Palacios, Chiara Mozzetta, Vincenzo Costanzo, Fabrizio d'Adda di Fagagna, Judith Hariprakash, Koustav Pal, Paolo Maiuri, and Mattia Forcato for critical feedback on the manuscript. We are grateful to Mariangela Panetta, Elisa Cesarini, Tom Misteli, Roland Foisner, Giannino del Sal, Francesco Napoletano, and Valentina Saccone for providing support with cell cultures. We are grateful to Chiara Cordiglieri for the precious help at the confocal microscope and Luca Madaro for providing support in image processing during the revision. We thank Elisa Salviato for advice on statistical analyses and Judith Hariprakash for advice on the definition of active enhancers. We gratefully acknowledge the Progeria Research Foundation for providing primary human fibroblasts of HGPS patients. The following cell lines were obtained from the NIGMS Human Genetic Cell Repository at the Coriell Institute for Medical Research: AG08498, AG07095. This work was supported by grants from the flagship CNR projects, (Epigen and Interomics) to C.L. and G.O, My First AIRC Grant (MFAG) n. 18535 to C.L., AFM-Telethon n. 21030 to C.L. and F.F.; by AIRC Start-up grant 2015 n.16841 to F.F.; and Cariplo 2017-0649 to C.L. and F.F. E.S. was supported by the Structured International Post-doc Program of SEMM (SIPOD) and the AFM-TELETHON fellowship n. 21835.

## Author contributions

In our opinion, F.L and C.P. equally contributed to this work and should be both considered as second authors. C.L. designed and F.M., F.L., A.B., and I.O. performed the experiments. E.S., C.P., and F.F. analyzed all sequencing data. S.V. contributed to analyze ChIP-seq data. L.A, F.G., and G.O. performed image processing and analysis. E.S., F.M., C.L., and F.F. interpreted the results and wrote the manuscript. All authors reviewed and approved the manuscript for submission.

## Competing interests

E.S., F.M., F.L., F.F., and C.L. declare a patent application has been submitted for the SAMMY-seq technique, thus constituting a potential competing interest. The remaining authors declare no competing interests.
