## [Peer Review File · Nature Communications]

Reviewers' comments:

Reviewer #1 (Remarks to the Author):

In "A new high-throughput sequencing-based technology reveals early deregulation of bivalent genes in Hutchinson-Gilford Progeria Syndrome" Sebestyen et al introduce an approach comparing sequentially digested fractions of chromatin by sequencing and argue the differences provide information about accessibility with their S4 most protected fraction largely overlapping with lamina signal. The value in this approach is that it avoids the cross-linking and transfection required by other approaches, both of which could potentially interfere with results and the latter of which reduces efficacy in tissue and primary cells. This is particularly important because the vast majority of work published in genome organization and chromatin accessibility uses cancer cell lines that may not share the total 4D genome organization found in tissues. They then apply their approach to investigate differences between control primary fibroblasts and progeria fibroblasts, finding that there are notable differences between the controls and progeria condition although there were not particularly consistent changes characterizing the disease state other than many changing genes being involved in stem cells, cancer and polycomb regulation. Notably, using early stage passages of the progeria cells they note that changes in accessibility could be detected prior to the loss of H3K9me3 that has been more focused on by the field and their findings suggest that polycomb changes are more important at least for the earlier progression of the disease. The data regarding the epigenetic changes and role of polycomb is solid and this finding is important for the field, especially with all the previous hype in journals such as Science and PNAS about the importance of H3K9me3 loss in progeria progression. Re the SAMMY-Seq method, it is mostly well described; however, there are a few details that I could not find and additional comparisons I would like to see in the data that are important for readers to appreciate the potential value of the approach.

First, to better appreciate the strength of the differences and similarities observed it is important to give information about the actual passage number of the cells used and ages of both controls and patients when samples were taken and I cannot find this information. Others who might use this approach will need to know if they should be expecting similarity in age-matched or non-age-matched samples so that they can assess what might have gone wrong if they don't get similar levels of overlap in their controls.

Second, while it is clear from the figure legends what is being shown in the figures, it is not clear if their overlap numbers given in the text and their Jaccard Index correlation analysis reflects data obtained for just the one control being shown in the figure or for averaged data from all three controls. Information about how calculations were done is clearly given in the methods, but the above question of exactly what samples were compared is not clear from either the text or the methods. In general, the high percentage of the genome identified in each fraction and amount of overlap could indicate either procedural variability in the extraction, differences in accessibility for extraction related to variations in 3D genome organization, and/or high backgrounds. This makes it important to see more statistical analysis of the differences between samples at multiple levels. We are given the average % of the genome covered in the S2, 3 and 4 fractions, but it is even more important to know for example if the 58% for S2 was 57%, 58% and 59% in each of the three control fibroblast lines or 48%, 58% and 68% and likewise the variance in the other fractions. So I would like to know how well does the subtracted pattern of the averaged three replicates reproduce when each individual S4 vs S2 run is compared? In the methods it states simply that "we calculated the genome wide differential signal for all comparisons, using the SPP package (version 1.15.4)" so I would assume the data is already calculated and it should not be much work for either this to be clarified in the text or for additional data using different comparisons to be shown. I think this is particularly important since the one place where we see a comparison of the three controls in Figure 2b shows lamin A overlap ranging from 47 to 60% and lamin B overlap from 16 to 35% and this is a reasonable amount of variation. H3K9me3 levels are also almost 2-fold difference between control samples.

Third, it would also be interesting to plot the variance over different chromosomes and relate this to information published by Joanna Bridger's laboratory about the positioning of these

chromosomes with respect to the periphery in progeria patients versus controls.

Fourth, where it is noted on page 7 of the pdf that lamin A overlap with SAMMY-Seq domains was 79% on average and then saying the A is 21% higher than the B. This phrasing is unclear. If they mean that it is actually 21% higher than what the B signal was then the B would be 65% whereas they could also mean that there was 79% overlap with lamin A and 58% overlap with lamin B. Also, the same question applies as for the second point and moreover, it would be good to know how much overlap there is specifically with the S2, S3 and S4 fractions. Moreover, in terms of explaining this section to the reader, they should clarify not only that the data they are using is for lamin B1 (they just state lamin B in the text so it could have been either B1 or B2), but also that it was taken from a study of cellular senescence whereas the lamin A data was actually taken from a progeria study. Thus, one might question could the lamin A data have more overlap because they are from a closer cell system? However, there are other possible interpretations of these numbers that depend on aspects of how analysis was done that remain unclear. For example, in figure 2 the trace shown has a peak in ATAC-Seq and DNase treatments for accessibility where there is a sharp drop in lamin A and B signals. At this same place the Sammy-Seq drops. However, the lamin A signal just drops sharply where the ATAC-Seq peak is and then comes back up while both the lamin B and Sammy-Seq signals stay down. Also towards the end of the shown trace there is a peak in B that is also paralleled by the Sammy-Seq where there is just a weak unchanging signal for lamin A. Thus visually, the Sammy-Seq trace seems to follow the lamin B1 trace better than the lamin A trace even though the number is given that there is much more overlap with lamin A. This raises the question if the numbers given are from a binary above or below the line as opposed to taking intensity also into account or matching places with signals shift. It would be interesting to analyze the data all three ways and such information would be useful to the readers and so I feel should be added. With respect to this lamin A/B1 difference clarifying this point is very important as it makes a big difference for interpretation of the results for understanding also the lamina role in genome organization and architecture if there is more tracking with the peripheral genome (lamin B1) or with lamins as structural networks irrespective of whether they are in the nuclear interior or periphery (lamin A).

Finally, I should highlight here that I think the work has value regardless of the outcome of the above suggested analysis both for the clear epigenetics results and for the technique itself, but I also think it is important to show this additional analysis of the SAMMY-Seq data and give the details so that a reader can better judge whether this approach is more likely to be of use to them for their particular system.

Reviewer #2 (Remarks to the Author):

In the manuscript "A new high-throughput sequencing-based technology reveals early deregulation of bivalent genes in Hutchinson-Gilford Progeria Syndrome", Sebestyén et al. developed a new high-throughput sequencing-based method, named SAMMY-seq, for genome-wide characterisation of changes in heterochromatin accessibility. Application of SAMMY-seq to primary fibroblasts derived from Hutchinson-Gilford Progeria (HGP) patients identified early alterations of chromatin structure that affect the expression of PcG regulated bivalent genes and precede changes in H3K9me3 marks deposition.

There is great need for the development of novel technologies to investigate the role of heterochromatin in a genome-wide fashion and the type of data generated by SAMMY-seq can be potentially very interesting for future applications. However, authors have not gone in great depth to characterise different aspects of the technology and assess advantages and limitations of SAMMY-seq. Also, biological relevance for their finding needs to be improved as well. Finally, the manuscript needs to be edited as many claims are not well supported. I think the paper would be appropriate for publication in Nature Communications if the following aspects are addressed:

Major points:

1. The manuscript contains a fair amount of imprecision. Examples are below:

a) The authors mention that MAPQ above 0 is unique when using bwa alignment. This information is not correct as unique reads have a MAPQ= 37. I urge the authors to perform again the analyses using MAPQ= 37 when referring to uniquely mapped reads.

b) The authors mention in the Discussion "Currently, we are successfully applying SAMMY-seq to many other experimental systems such as human epithelial and myogenic cells, human lymphocyte subpopulations or murine muscle satellite stem cells, in some cases scaling down to 10.000 cells (data not shown)." Since SAMMY-seq data were produced by using 4.000.000 cells, sequencing data generated by using 2 orders of magnitude less cells represent a major technological advance and it would allow the study of more rare cell types. The authors need to support the sentence by showing how SAMMY-seq data looks when scaling down the number of input cells as mentioned in the Discussion.

c) I failed to find information about the passage number of fibroblasts used to generate data in Figure 3 and 4. As it is stated that differences in chromatin organisation are present in early passage cells, it is crucial to provide the information regarding passage number. This information will also help reviewers and readers to compare the amount of progerin present as shown in Supplementary Figure 3a.

2. I'm not convinced using S4 fraction translates into better information compared to S3 fraction. In Figure 1c, the differential reads distribution for S4 vs S2 and S3 vs S2 seem to be quantitative rather than qualitative. Can the authors provide some analyses that clearly show the advantage of S4 over S3? How much more information a user of the technology gets when including S4 as well?

3. Can the authors provide some mechanistic evidence for the early changes of chromatin structure in HGPS fibroblasts? For example, overexpression of progerin in control cells followed by SAMMY-seq analysis might recapitulate the profile observed in HGPS patients. Or would a knock-down of progerin in HGPS cells and cause some rescue in the SAMMY-seq profiles? This type of experiments would also widen the array of applications of the technology.

4. PLA assay shows that Lamin A interacts with PcG proteins in control or HGPS fibroblasts whereas progerin interacts with Ezh2 in late passage HGPS. However, it would be more convincing if the authors would provide evidence also with a biochemical assay, showing the same interactions by using immunoprecipitation assay as well.

5. The authors correctly mentioned that Hi-C experiments failed to detect alterations for 3D DNA contacts in early passage HGPS as they were evident only at late passage cells. SAMMY-seq detects reduced enrichment of S4 vs S2 in HGPS fibroblasts only, suggesting alterations in chromatin organisation in patients cells. How about SAMMY-seq signal at late-passage HGPS? Does the S4 vs S2 enrichment profile change? Is it completely lost?

6. The authors do not find alterations in H3K9me3 patterns when performing ChIP-seq using antibody against this specific histone modification. One possible explanation for this result is the use of formaldehyde to crosslink cells that dramatically decreases to signal-to-noise ratio. Recently, a new technology, Cleavage Under Targets and Release Under Nuclease (CUT&RUN, Skene et al, Elife 2017), has been published combining native conditions and antibody-directed MNase digestion to ensure that chromatin cleavage happens solely close to protein of interest. This technology does not require the use of crosslinking agents or extensive sonication, thus greatly reducing the signal from unwanted genomic regions. The authors should perform CUT&RUN using H3K9me3 antibody to convincingly show the absence of significant differences in H3K9me3 patterns between controls and HGP patients.

Minor:

- From the main text it is not clear which are the major differences between SAMMY-seq and previously published NGS methods that employ chromatin fractionation. Can the authors provide a table that compares across SAMMY-seq and other methods with regards to conditions, enzymes and cellular models? This would help both reviewers and readers to understand the advantages of SAMMY-seq over already existing NGS methods.
- I do not find surprising that correlation becomes more stable for genomic bins of larger size. Can the authors provide analyses to estimate the resolution of SAMMY-seq?
- The authors keep on changing the chromosomal regions shown in the Main Figures. What is the rationale for this as it seems rather confusing. I suggest to use same chromosomal regions or at least give the rationale why authors constantly change the chromosomal area.
- Can the authors speculate why S2 fraction is more variable across biological replicates? Is it somehow related to the digestion time of DNase?
- Related to major point 2: it is needed to show also correlation for S3 vs S2, S4 vs S2 and S3 vs S4 to establish how different the fractions are with each other.
- Also related to major point 2: it would be very useful to perform the same analyses depicted in Supplementary Figure 2c using S3 vs S2 fractions.

Reviewer #3 (Remarks to the Author):

In this manuscript, the authors developed a high-throughput sequencing-based method, SAMMY-seq to detect the genome-wide heterochromatin accessibilities in primary cells. With this approach, the authors first characterized the chromatin accessibility changes in early-passage fibroblasts derived from HGPS patients, and their correlations with dysregulated H3K27me3 marks and Polycomb bivalent genes expression. They drew the conclusion that "chromatin structural changes are early events in HGPS nuclear remodeling and interfere with proper PcG control". This work provided new insights into understanding the early-onset heterochromatin remodeling events that contribute to premature senescence in HGPS fibroblasts. Outlined below are specific comments to help improve this manuscript.

Major Issues

- In this study, SAMMY-seq enrichment was normalized with signals from less condensed fractions. However, S4/S2 vs S4/S3 sometimes show different trends (e.g. Fig 3c). Under circumstances of inconsistency, is there a criterion to determine which normalization is more reliable?
- Why are the SAMMY-seq regions preferentially overlap with the lamin A/C associated domains comparing to lamin B associated regions?
- What are the passage numbers of the control fibroblasts? In this study, the patterns of SAMMY-seq domains showed decent consistency among different control samples, will this change over passages/aging?
- Figure 3a: the SAMMY-seq domains appeared scattered in different HGPS samples. Is it possible to find an HGPS-specific pattern?
- Figure 5b: HGPS188 is not significant in the selected region. Only 2 controls are shown.

Minor Issues

- Figure 1c: "Regions of signal enrichment or depletion over the reference samples are marked in yellow or purple, respectively" Where are the marks?
- Figure 4e: how was the quantification performed. What were the sample sizes?
- Please fix typos

Reviewer #4 (Remarks to the Author):

By sequential extraction of soluble fractions under different conditions, Sebestyen et al. have developed a method to extract heterochromatin of increasing compaction. By applying this method on progeria patient samples, authors have determined that chromatin organization changes take place at the most compacted heterochromatic regions (loss of such organization) that precedes H3K9me3 changes with earlier passages of primary cell culturing. They have also determined changes appear to take place at H3K27me3 enriched regions, specifically transcriptional repression at the bivalent regions appear to be lost.

General comments:

Although chromatin changes with the progeria patient samples are interesting, it felt like it took a long time to reach these observations. The first half that describes SAMMY-seq was not very easy to read through, due to points mentioned below, as well as the writing that should be improved to help the readers follow with more ease.

One thing I could not help but wonder is how different SAMMY-seq is with other non-antibody-dependent methods to extract heterochromatin such as by different salt concentrations Steve Henikoff had used. If these data are available for the same cell types, comparisons to other methods would improve the manuscript, especially if it can extract regions that are not by other already available methods. Along this line, the novelty of this method is not very clear. If the most condensed heterochromatin is being extracted with this method which other methods do not, it should be more clearly stated. Although this aspect is mentioned in the Discussion, I do not see data in this manuscript that supports the statement 'SAMMY-seq represents a significant improvement in the field of chromatin characterization as it provides novel information complementary and not overlapping with other high-throughput sequencing based methods commonly used to study chromatin structure and function', since comparative analysis with other methods have not been carried out in this study.

Although the authors did not see differential gene expression at the protein coding genes, I wonder if expression of enhancer RNAs are affected at all. This may be outside the scope of this manuscript but something to consider for in the future.

I find the conclusions confusing at the end, since although changes are observed at the most condensed heterochromatin (S4 vs S2), the changes that are discussed are in the bivalent regions, which are not the most condensed heterochromatin. The logic or the model should be explained better.

It would help the readers to have the histone modification antibody target name by the browser shots in the figures.

Specific comments:

In the third line of the Results, it is written 'DNase-sensitive chromatin (S2 fraction), ...'. Is this a typo for 'insensitive'? Obviously the S2 fraction does not contain DNase-sensitive chromatin, evident from the browser shot profiles of the S2 fractions. If it is DNase-sensitive, it should look like DNase-seq and ATAC-seq, which it does not. I believe it contains the chromatin that was not released by DNase. It took some time for me to figure this out at the beginning and I was very confused reading the manuscript especially when looking at the figures. This needs to be addressed in the text. Also although it is described in the figures, some descriptions of how sequential extraction was carried out should be mentioned in the text such as using urea for S4 fractions.

Page 6: 7th line of results: 'We first applied SAMMY-seq to 3 independent normal skin primary fibroblast cell lines, ...'. Instead of using the word 'normal', 'healthy' should be used (e.g. 3 independent normal skin primary fibroblast cell lines, originating from 3 different healthy individuals'

Page 6: 'At 1Mb resolution, the mean Spearman correlation is 0.27 for the S2 fraction, 0.92 for S3, and 0.79 for S4.': correlation between what? Controls? If so, why is S4 vs S2 more consistent? If S2 fraction is less reproducible, it's not clear to me why S4 vs S2 would result in more conserved.

Page 7: 'Of note, other chromatin fractionation-based NGS methods have been previously described^{41,42}, however they are not directly comparable to our technology as they adopt different conditions, enzymes and cellular models. ': this should be mentioned much earlier. In addition, other methods that detect heterochromatin by non-antibody-dependent approaches should be mentioned a little bit in more details than it is currently. For example, Becker et al., Mol. Cell 2017 and Nicetto, Science 2019.

Page 7: 'Using data from the Roadmap Epigenomics consortium⁴³ and other publications^{29,44,45} we noticed that SAMMY-seq signal (S4 vs S2 enrichment) is highly consistent across biological replicates (CTRL002, CTRL004, CTRL013), ': I find this statement strange. Consistency between replicates do not requires other datasets (such as the Roadmap).

Page 7: 'Interestingly, the overlap with Lamin A/C associated domains was 21% higher than with Lamin B associated domains, suggesting that SAMMY-seq preferentially enriches for regions interacting with Lamin A/C. ': not sure if this can be stated, since it can be an antibody quality issue. Furthermore, if the used chromatin was prepared in a standard manner (as it is mentioned in the methods), the most condensed chromatin that is observed in the S4 fractions may not be contained in this ChIP experiment, therefore would not see these regions being immunoprecipitated.

Page 10: 'To further investigate PcG role in early passage HGPS fibroblasts, we examined the total amounts of PRC1-subunit Bmi1 and PRC2- subunit Ezh2 (Supplementary Figure 6a, b), ': should be mentioned in the text that this is at the protein level.

Page 11: 'By visual inspection we noticed that while the main peaks were present in both controls and HGPS samples, H3K27me3 signal was spread over flanking regions in HGPS (Figure 5b).': this should be quantitated (amount of the genome enriched by H3K27me3).

Page 11: 'These findings overall suggest that early chromatin remodelling has an impact on a subset of PcG targets, the bivalent genes, more susceptible to variations of PcG occupancy⁴⁹ (Figure 6d). ': number of bivalent genes that are upregulated should be mentioned.

Page 11/12: 'supporting the previous observation that PcG regulated regions shift towards the insoluble S4 fraction in HGPS (Figure 6b). ': Not sure how this is concluded from 6b. I am also confused as to how Progerin-PcG crosstalk hypothesis was formed from the data.

Figure 2: all the browser shots are so zoomed out. I'd like to see how they look when it is zoomed in more, are they continuous for a long stretch like H3K9me3? At this resolution, it looks like H3K27me3 is overlapping with H3K4me1 and open regions (ATAC and DNase), which should not be unless they are bivalent regions. Also, H3K4me1 track should be directly below ATAC/DNase tracks, bundling the active mark with the open regions.

Figure 3a: Does the amount/number of SAMMY-seq domains change in progeria samples? Is it simple loss or are they acquired at all elsewhere? From the browser shot in figure 3a, it looks like there is a global loss but there seems to be acquisition in surrounding regions. Of particular notice, HGPS188 seems to acquire a large region. This should be mentioned in the text and whether general acquisition is observed should be analyzed. In such analysis, it may also be interesting to determine whether the acquired SAMMY-seq regions are as large as the controls. Additionally, it would also be nice to see if chromatin accessibility changes at all with the loss of SAMMY (S4 vs S2) domains (such as with ATAC-seq).

In Figure 3c, S4 vs S2, two patients clearly have different patterns from the control samples, however, one patient shows similar profiles as the control samples. The patient sample numbers should be shown here so that it can be compared with other data such as the browser shots.

Figure 5b: would be good to see the SAMMY-seq data as well in the browser shot to see if it coincides with the loss of SAMMY-seq regions.

Figure 6a: In terms of the story flow (the H3K27me3 story) and placement in the text (at the end of the paragraph that describes figure 5), this figure belongs to Figure 5, although 'spreading' and 'loss' are not very consistent. This figure is also missing the legend on the y-axis. The number of genes being analyzed here should also be mentioned in the text at least.

Figure 6d: the text only mentions about SAMMY-seq domains, but the figure is of H3K27me3 ChIP. This is very confusing and hard to follow logics. Additionally, it would be good to have the H3K4me3 ChIP-seq track to show bivalency.

Minor comments

- Figure 2a: it doesn't break the manuscript, but ATAC-seq and DNase-seq are redundant, both of which assay chromatin accessibility

- Figure 2a: 'H3' should be added before the K9me3 etc, consistent with other figures (and be accurate nomenclature-wise)

Figure 3a: 'K9me3' should be 'H3K9me3' like elsewhere

- Figure 3a: colors of lamin A/C and lamin B should be changed so that it is easier for the readers to distinguish data between these ChIP and SAMMY-seq.

- for all browser shots, if the antibody used to ChIP is shown on the left, it would be helpful for the readers

We thank the reviewers for their constructive feedback.

Following the reviewers' comments, the additional data and analyses allowed us to refine the conclusions presented in our manuscript. In particular:

- We confirm that early passage HGPS cells show alterations of chromatin accessibility in the normally heterochromatic domains associated to nuclear lamina.
- We confirm that the early molecular alterations in HGPS cells involve the crosstalk between Lamina and Polycomb regulation. The altered crosstalk in Progeria that especially affects the bivalent gene regulation is now supported by more quantitative and statistical analyses.
- We extended our comparative analyses to a larger set of published ChIP-seq datasets, to analyse the correlation between SAMMY domains and Lamin A or Lamin B1 LADs. We do not find a specific marked difference in correlation between Lamin A/C and Lamin B1
- We reprocessed all ChIP-seq datasets with the same analysis pipeline in order to have a more uniform treatment of public and proprietary ChIP-seq datasets. The amended figures actually confirm an even clearer correspondence between SAMMY-seq and Lamin or H3K9me3 ChIP-seq profiles.
- We show new data in support of SAMMY-seq reliability and reproducibility, also when applied to as little as 10,000 cells. Moreover, if needed we may also show data on other cell types. However, so far we have not added such data to the revised manuscript as we deemed this out of scope.
- We added two tables to more clearly illustrate the differences between SAMMY-seq and:
 - o *Other genome-wide methods to study LADs and heterochromatin structure*
 - o *Other genome-wide methods based on salt-extracted chromatin fractions*
- We thoroughly revised the text and we amended the figures with labels and colours to improve readability

For more details about the individual comments, please review the point by point reply below.

Reviewer #1 (Remarks to the Author):

In "A new high-throughput sequencing-based technology reveals early deregulation of bivalent genes in Hutchinson-Gilford Progeria Syndrome" Sebestyén et al introduce an approach comparing sequentially digested fractions of chromatin by sequencing and argue the differences provide information about accessibility with their S4 most protected fraction largely overlapping with lamina signal. The value in this approach is that it avoids the cross-linking and transfection required by other approaches, both of which could potentially interfere with results and the latter of which reduces efficacy in tissue and primary cells. This is particularly important because the vast majority of work published in genome organization and chromatin accessibility uses cancer cell lines that may not share the total 4D genome organization found in tissues. They then apply their approach to investigate differences between control primary fibroblasts and progeria fibroblasts, finding that there are notable differences between the controls and progeria condition although there were not particularly consistent changes characterizing the disease state other than many changing genes being involved in stem cells, cancer and polycomb regulation. Notably, using early stage passages of the progeria cells they note that changes in accessibility could be detected prior to the loss of H3K9me3 that has been more focused on by the field and their findings suggest that polycomb changes are more important at least for the earlier progression of the disease. The data regarding the epigenetic changes and role of polycomb is solid and this finding is important for the field, especially with all the previous hype in journals such as Science and PNAS about the importance of H3K9me3 loss in progeria progression. Re the SAMMY-Seq method, it is mostly well described; however, there are a few details that I could not find and additional comparisons I would like to see in the data that are important for readers to appreciate the potential value of the approach.

We thank the reviewer for his/her constructive feedback. We addressed each specific comment as detailed below:

First, to better appreciate the strength of the differences and similarities observed it is important to give information about the actual passage number of the cells used and ages of both controls and patients when samples were taken and I cannot find this information. Others who might use this approach will need to know if they should be expecting similarity in age-matched or non-age-matched samples so that they can assess what might have gone wrong if they don't get similar levels of overlap in their controls.

We agree with the reviewer that indeed the passage number is a crucial piece of information, especially when dealing with Hutchinson-Gilford Progeria Syndrome (HGPS) cell lines.

In the original submitted version of the manuscript we indicated, in the results section that we considered as "early-passage" cells with passage number between 10 and 12. Quoting from the original text:

"We applied SAMMY-seq to investigate chromatin changes in early-passage skin fibroblasts from 3 independent HGPS patients (passage number from 10 to 12)"

We also indicated details of passage number for cells used in specific figure panels, marked with the letter "p" followed by the passage number (e.g. "p11" for passage 11): Figures 4d, 6d-f, and Supplementary figures 4 (all panels), 5c and 7 panels a,c,e-h.

To better clarify the passage number used in each experiment, we revised the manuscript to make sure that the exact passage number is cited in each figure panel and legend where appropriate.

As suggested by the reviewer, we also added the details for the ages of all controls and patients. These values are reported in the Methods section in the "Cell culture" paragraph.

Second, while it is clear from the figure legends what is being shown in the figures, it is not clear if their overlap numbers given in the text and their Jaccard Index correlation analysis reflects data obtained for just the one control being shown in the figure or for averaged data from all three controls. Information about how calculations were done is clearly given in the methods, but the above question of exactly what samples were compared is not clear from either the text or the methods. In general, the high percentage of the genome identified in each fraction and amount of overlap could indicate either procedural

variability in the extraction, differences in accessibility for extraction related to variations in 3D genome organization, and/or high backgrounds. This makes it important to see more statistical analysis of the differences between samples at multiple levels.

We are given the average % of the genome covered in the S2, 3 and 4 fractions, but it is even more important to know for example if the 58% for S2 was 57%, 58% and 59% in each of the three control fibroblast lines or 48%, 58% and 68% and likewise the variance in the other fractions.

We agree this is an important point. The genome covered in individual samples, as requested by the reviewer, was reported in the form of a bar plot in Supplementary Figure 1c in the original manuscript. To make these details more clearly readable, we amended the bar plot in Supplementary Figure 1c by explicitly indicating the number for the value reported in each bar.

So I would like to know how well does the subtracted pattern of the averaged three replicates reproduce when each individual S4 vs S2 run is compared? In the methods it states simply that "we calculated the genome wide differential signal for all comparisons, using the SPP package (version 1.15.4)" so I would assume the data is already calculated and it should not be much work for either this to be clarified in the text or for additional data using different comparisons to be shown. I think this is particularly important since the one place where we see a comparison of the three controls in Figure 2b shows lamin A overlap ranging from 47 to 60% and lamin B overlap from 16 to 35% and this is a reasonable amount of variation. H3K9me3 levels are also almost 2-fold difference between control samples.

Based on this comment we understand there was an unclear explanation of the analysis reported in Figure 2b. This figure is not reporting an overlap between SAMMY-seq samples and chromatin marks, instead the figure is reporting the "Kernel correlation" as stated in the main text

"We confirmed this is a consistent genome-wide pattern using StereoGene kernel correlation, a method for the unbiased comparison of different types of chromatin marks (Figure 2b and Supplementary Figure 2a)"

and in the respective figure legend:

"Genome-wide kernel correlation calculated by StereoGene, between SAMMY-seq fraction comparisons in individual control samples against ChIP-seq and other chromatin marks. These include: ATAC-seq, DNase-seq, Lamin A/C and Lamin B1, H3K27me3, H3K4me1, H3K9me3, H3K36me3, H3K27ac and H3K4me3."

Instead Supplementary Figure 2c reports the actual overlap between significantly enriched SAMMY-seq domains (as defined by the EDD algorithm) vs the significantly enriched H3K9me3 domains. In this case the variability is not 2-fold as instead it ranges between 37% and 49% (for S4 vs S2). To make these details more clearly readable, we amended the bar plot in Supplementary Figure 2c by explicitly indicating the number for the value reported in each bar. It's worth remarking that Supplementary Figure 2c has been further extended by adding results for S4 vs S3 and S3 vs S2 comparisons to address another reviewer's comment.

In principle we do not find surprising some level of variability across different controls as these are primary fibroblasts derived from different genetically unrelated individuals.

Third, it would also be interesting to plot the variance over different chromosomes and relate this to information published by Joanna Bridger's laboratory about the positioning of these chromosomes with respect to the periphery in progeria patients versus controls.

The reviewer's comment is addressing an interesting point about the direct correlation between SAMMY-seq domains and chromosomes localization at the nuclear periphery. However, heterochromatin is not exclusively located at the nuclear periphery according to literature and in line with our own data on H3K9me3 (Figure 4d).

However, as suggested by the reviewer, we examined the percentage of each chromosome length covered by SAMMY-seq domains (S4 vs S2 comparison). As reported in (Figure R1) below, there is large variability across chromosomes as well as across samples.

Figure R1: Chromosome by chromosome coverage in SAMMY-seq domains. The bar plots report for each sample (indicated on the right side label) the percentage (y-axis) of each chromosome (x-axis) that is covered by a SAMMY-seq domain (S4 vs S2 comparison). Individual values are reported inside each bar plots, close to the upper border of each plot.

Then, we also considered Joanna Bridger's articles showing FISH data on the preferential distribution of chromosomes with respect to the nuclear periphery. In particular, we reviewed data in (Mehta *et al.*,

Genome Biology 2011) and (Bikkul *et al.*, Biogerontology 2018). In these articles, all results are actually focused only on chromosomes 10, 18 and X.

We excluded chromosome X to avoid biases related to different sexes. Instead, in control cells, chromosome 10 and 18 were reported to be prevalently in the nuclear center and periphery, respectively.

We also considered chromosome 19, because it is similar in length to chromosome 18. Moreover, both chromosome 18 and 19 are reported to have an evolutionarily conserved positioning across primates (Tanabe *et al.*, PNAS 2012), with preferential location in the nuclear periphery and nuclear center, respectively.

We can see in the genomics data tracks that the more peripheric chromosome 18 has generally more SAMMY-seq domains (S4 vs S2) than the centrally located chromosome 19 (see Figure R2 below).

Figure R2: SAMMY-seq in peripheral and central chromosomes. The figure shows genomics tracks for SAMMY-seq reads enrichment in S4 vs S2 fractions comparison for chromosome 18 and 19, which have preferential location in the nuclear periphery and nuclear center, respectively. We can see in the genomics data tracks that the more peripheric chromosome 18 has generally more SAMMY-seq domains, marked with grey rectangles (S4 vs S2), than the centrally located chromosome 19.

It's also worth mentioning that high-throughput sequencing based assays do not achieve complete genome coverage due to technical limitations. For example, we generally discard multimapping reads, as well as blacklisted regions with low sequence complexity, including telomeric and centromeric regions, as described in Methods section. These regions are instead generally well covered by 3D FISH based assays.

We verified that the peripheral chromosome 18 has in general a trend towards higher percentage of (S4 vs S2) SAMMY-seq domains in control samples, with respect to progeria samples (see Figure R3 below). On the contrary, the centrally located chromosomes 10 and 19 have a generally lower percentage of (S4 vs S2) SAMMY-seq domains in control samples rather than in progeria samples. However, these differences are not statistically significant, due to the large inter-individual variability (see Figure R3 below). Thus we would prefer not to make a point on this result in the revised manuscript.

Figure R3: SAMMY-seq domains coverage in peripheral and central chromosomes. The plots report the coverage (y-axis, percentage of chromosome length) of SAMMY-seq domains (S4 vs S2) for chromosome 18 (preferentially located at the nuclear periphery) and chromosomes 10 and 19 (preferentially located in the nuclear center). The dots represent the coverage value for each control (CTRL) or progeria (HGPS) sample. The horizontal lines mark the average and the whiskers show the standard error over mean.

Fourth, where it is noted on page 7 of the pdf that lamin A overlap with SAMMY-Seq domains was 79% on average and then saying the A is 21% higher than the B. This phrasing is unclear. If they mean that it is actually 21% higher than what the B signal was then the B would be 65% whereas they could also mean that there was 79% overlap with lamin A and 58% overlap with lamin B.

As suggested by this and other reviewers, we revised the analysis comparing SAMMY-seq domains with Lamin A/C or Lamin B1 ChIP-seq data. The revised manuscript includes a more comprehensive comparison with multiple datasets for Lamin ChIP-seq, comprising data from Lund *et al.* Nucleus 2015; McCord *et al.* Genome Res. 2013; Lund *et al.* Nucleic Acids Res. 2014; Dou *et al.* Nature 2015; and Sadaie *et al.* Genes Dev. 2013. The new results are summarized in the revised Figure 2 and Supplementary Figure 2. The difference in association to Lamin A/C rather than Lamin B1 is milder when considering a comprehensive set of ChIP-seq profiles.

The specific sentence cited by the reviewer in this comment has been updated as follows:

“Despite some inter-dataset variability across a comprehensive set of independent Lamin ChIP-seq profiles (Lamin A/C and Lamin B1) for human fibroblasts, both Lamin A/C and Lamin B1 show similar correlation values (Supplementary Figure 2a), thus only their average is summarized in (Figure 2b).”

Then later:

“When comparing SAMMY-seq domains with Lamin A/C LADs, we observed an average 0.49 Jaccard Index overlap, when considering all samples and fractions comparisons (Supplementary Table 4). Each of these pairwise comparisons has a significantly high overlap, as assessed by randomizing either LADs or SAMMY-seq domains (Figure 2c and Supplementary Figure 2b)”

Also, the same question applies as for the second point and moreover, it would be good to know how much overlap there is specifically with the S2, S3 and S4 fractions.

On the base of this comment we understand there was maybe a misunderstanding on some analysis details. Namely, we identify SAMMY-seq domains by applying the EDD algorithm to the comparison between distinct fractions. Thus the SAMMY-seq domains are genomic coordinates intervals that can be compared to “peaks” or “enrichment regions” identified by ChIP-seq on other chromatin marks. This comparison is based on an “overlap” between genomic coordinate intervals. This is the case of the comparisons reported in Supplementary Table 4.

However, here the reviewer is referring to “overlap” when mentioning individual fractions (S2, S3 and S4 fractions). We never define “domains” based on single fractions. Instead, we can compare individual SAMMY-seq fractions with chromatin marks by computing a “correlation” between their genomic tracks. This is the analysis was reported in the original manuscript in Supplementary Figure 2a.

However, in the revised version we decided to substitute this supplementary Figure 2a with an extensive correlation of several independent datasets with all SAMMY comparisons (S4 vs S2; S4 vs S3; S3 vs S2) (New Supplementary Figure 2a). For a better readability of the manuscript we decided to remove the Figure showing the correlation with the individual fractions (S4, S3 and S2 considered individually) that we present here for completeness and transparency.

Figure R4: Genome-wide kernel correlation calculated by StereoGene, between read coverage in individual control samples chromatin fractions against ChIP-seq and other enrichment signals in ATAC-seq, DNase-seq, Lamin A/C and Lamin B; H3K27me3, H3K4me1 and H3K9me3; H3K36me3, H3K27ac and H3K4me3. In most samples the correlation is progressively increasing for closed chromatin marks or decreasing for open chromatin marks when considering fractions from S2 to S4.

Moreover, in terms of explaining this section to the reader, they should clarify not only that the data they are using is for lamin B1 (they just state lamin B in the text so it could have been either B1 or B2),

We thank the reviewer for pointing out this oversight. We have now revised Figure 2 and figures reporting other related analyses to explicitly state that it is using Lamin B1 ChIP-seq data. Accordingly, we also amended the manuscript text in all of the references to the same data.

but also that it was taken from a study of cellular senescence whereas the lamin A data was actually taken from a progeria study. Thus, one might question could the lamin A data have more overlap because they are from a closer cell system?

Following this important point raised by the reviewer, we analysed a comprehensive set of published ChIP-seq datasets for Lamin A/C and Lamin B1 in human fibroblasts. As detailed in the new Supplementary Figure 2a, the correlation with SAMMY-seq samples is similar for Lamin A/C and B1, despite some variability across individual samples. We summarized these results in the revised main Figures 2b by reporting the average across all of these public datasets. We also updated the Figure 2c, Supplementary Figures 2b and 2c, by using a consensus set of Lamin A/C or Lamin B1 LADs: i.e. domains identified in at least 2 Lamin A/C or Lamin B1 ChIP-seq samples, respectively.

Based on these results, the inter-dataset variability in Lamin A/C or Lamin B1 does not allow us to make a specific statement about the differences between the two lamins or between distinct cell systems.

Moreover, as pointed out also by reviewer #4, the comparison with different lamins ChIP-seq experiments may be hampered by the differences in immunoprecipitation efficiency for different antibodies.

However, there are other possible interpretations of these numbers that depend on aspects of how analysis was done that remain unclear. For example, in figure 2 the trace shown has a peak in ATAC-Seq and DNase treatments for accessibility where there is a sharp drop in lamin A and B signals. At this same place the Sammy-Seq drops. However, the lamin A signal just drops sharply where the ATAC-Seq peak is and then comes back up while both the lamin B and Sammy-Seq signals stay down. Also towards the end of the shown trace there is a peak in B that is also paralleled by the Sammy-Seq where there is just a weak unchanging signal for lamin A. Thus visually, the Sammy-Seq trace seems to follow the lamin B1 trace better than the lamin A trace even though the number is given that there is much more overlap with lamin A. This raises the question if the numbers given are from a binary above or below the line as opposed to taking intensity also into account or matching places with signals shift. It would be interesting to analyze the data all three ways and such information would be useful to the readers and so I feel should be added.

We would like to remark that thanks to this reviewer's comment, we extended our comparative analyses to a larger set of published ChIP-seq datasets, thus confirming that there is not a specific marked difference in correlation between Lamin A/C and Lamin B1, as reported in the novel Figures 2b-c and Supplementary Figure 2.

We also agree with the reviewer's opinion that it's important to examine concordance between different epigenomic datasets in terms of overlap between discrete peak calls (binary comparison, as defined by the reviewer), as well as in terms of magnitude of enrichment (intensity comparison, as defined by the reviewer), also accounting for minor shifts in signal position (as pointed out by the reviewer). However, we would like to remark that we already performed these alternative analyses, namely when we examine *a*) Jaccard Index overlaps between peak calls (binary comparison) (see novel Figure 2c, Supplementary Figure 2b) and *b*) when we report results based on Stereogene Kernel correlation (see novel Figure 2b, Supplementary Figure 2a), which accounts for intensity of the signal, while also allowing some possible minor shifts in the profiles, by using a spatially smoothed correlation of signals.

Finally, it must be noted that we updated the plots containing ChIP-seq tracks (Figure 2a, 3a and 4a), as well as related analyses, compared to the initial version. In the previous manuscript version we used the pre-processed data as published by the original authors. However, in order to have a more uniform

treatment of public and proprietary ChIP-seq datasets, we decided to reprocess all ChIP-seq datasets with the same analysis pipeline. We deemed the novel uniform pre-processing more appropriate for the comparison across datasets. This updated uniform processing of ChIP-seq didn't change the results in terms of statistical significance and quantitative analyses. However, the amended figures actually confirm an even clearer correspondence between SAMMY-seq and Lamin or H3K9me3 ChIP-seq profiles.

With respect to this lamin A/B1 difference clarifying this point is very important as it makes a big difference for interpretation of the results for understanding also the lamina role in genome organization and architecture if there is more tracking with the peripheral genome (lamin B1) or with lamins as structural networks irrespective of whether they are in the nuclear interior or periphery (lamin A).

Finally, I should highlight here that I think the work has value regardless of the outcome of the above suggested analysis both for the clear epigenetics results and for the technique itself, but I also think it is important to show this additional analysis of the SAMMY-Seq data and give the details so that a reader can better judge whether this approach is more likely to be of use to them for their particular system.

Reviewer #2 (Remarks to the Author):

In the manuscript “A new high-throughput sequencing-based technology reveals early deregulation of bivalent genes in Hutchinson-Gilford Progeria Syndrome”, Sebestyén et al. developed a new high-throughput sequencing-based method, named SAMMY-seq, for genome-wide characterisation of changes in heterochromatin accessibility. Application of SAMMY-seq to primary fibroblasts derived from Hutchinson-Gilford Progeria (HGP) patients identified early alterations of chromatin structure that affect the expression of PcG regulated bivalent genes and precede changes in H3K9me3 marks deposition. There is great need for the development of novel technologies to investigate the role of heterochromatin in a genome-wide fashion and the type of data generated by SAMMY-seq can be potentially very interesting for future applications. However, authors have not gone in great depth to characterise different aspects of the technology and assess advantages and limitations of SAMMY-seq. Also, biological relevance for their finding needs to be improved as well. Finally, the manuscript needs to be edited as many claims are not well supported. I think the paper would be appropriate for publication in Nature Communications if the following aspects are addressed:

Following the specific comments by this and other reviewers, the manuscript has been extensively revised. Most notably, related to the points above:

- 1) we added two tables to more clearly illustrate the differences between SAMMY-seq and:
 - a) *Other genome-wide methods to study LADs and heterochromatin structure*: including ChIP-seq, DamID, gradient-seq (Becker *et al.*, Mol Cell 2017; Nicetto *et al.*, Science 2019) and protect-seq (Spracklin and Pradhan, Nucleic Acids Res. 2020) (Supplementary Table 5).
 - b) *Other genome-wide methods based on salt-extracted chromatin fractions*: including Salt fractions profiling (Henikoff *et al.*, Gen Res 2009) and HRS-seq (Baudement *et al.*, Gen Res 2018) (Supplementary Table 1).
- 2) We added new data to prove SAMMY-seq flexibility to be adopted also a small number of cells: scale-down to 10,000 primary fibroblasts (See also the novel Supplementary Figure 3)
- 3) We revised and extended the description and comparison of our method advantages compared to other methods
- 4) We addressed the specific comments raised by this and other reviewers about individual biological claims
- 5) We extensively revised the manuscript to improve clarity and readability (see changes marked in red in the revised manuscript).

Major points:

1. *The manuscript contains a fair amount of imprecision. Examples are below:*
 - a) *The authors mention that MAPQ above 0 is unique when using bwa alignment. This information is not correct as unique reads have a MAPQ= 37. I urge the authors to perform again the analyses using MAPQ= 37 when referring to uniquely mapped reads.*

We thank the reviewer for this comment. We are aware of the difficulties to precisely define a MAPQ score threshold corresponding to uniquely mapped reads.

We actually verified that the percentage of reads with a MAPQ value greater than zero and lower than 37 ranges from 5% to 7% across all SAMMY-seq samples in our dataset (see also Figure R5 below). Thus discarding reads in this range of mapping quality would not change the results. We also amended the text in Methods section (paragraph “SAMMY-seq sequencing read analysis”) as well as the Supplementary Figure 1 legend, to correctly report these details.

Figure R5: Sequencing reads mapping quality distribution. Each bar plot in the figure shows the number of reads (million reads – y-axis) grouped by mapping quality score (MAPQ values – x-axis). For each control and progeria sample in the dataset (labels on the right hand side) and for each SAMMY-seq fraction (S2, S3 and S4 – see labels on top), the three bars report the percentage of reads with mapping quality 0, mapping quality between 1 and 36, or mapping quality 37, respectively.

b) *The authors mention in the Discussion “Currently, we are successfully applying SAMMY-seq to many other experimental systems such as human epithelial and myogenic cells, human lymphocyte subpopulations or murine muscle satellite stem cells, in some cases scaling down to 10.000 cells (data not shown).” Since SAMMY-seq data were produced by using 4.000.000 cells, sequencing data generated by using 2 orders of magnitude less cells represent a major technological advance and it would allow the study of more rare cell types. The authors need to support the sentence by showing how SAMMY-seq data looks when scaling down the number of input cells as mentioned in the Discussion.*

As the reviewer remarked, this point will be relevant for the broad applicability of SAMMY-seq technique. We have indeed performed SAMMY-seq on several cell types that will be the subject of independent publications. As such, those data are not pertaining to the current manuscript. However, to support the relevant claim that SAMMY-seq can be applied on small number of cells, we have performed scale-down experiments on control human primary fibroblasts. Namely, we applied SAMMY-seq on 250K, 50K and 10K cells. The scale-down experiments were performed in parallel by two distinct experimentalists, to further confirm the reproducibility and robustness of the technique. The results of the scale down experiment are now summarized in the novel Supplementary Figure 3.

Upon request, we may also share results on other cell types with the reviewers.

c) *I failed to find information about the passage number of fibroblasts used to generate data in Figure 3 and 4. As it is stated that differences in chromatin organisation are present in early passage cells, it is crucial to provide the information regarding passage number. This information will also help reviewers and readers to compare the amount of progerin present as shown in Supplementary Figure 3a.*

As explained for a similar point raised by other reviewers, we have now revised the manuscript and figures to make sure that the exact passage number is cited in each figure panel and legend. In particular, for Figure 3 the passage number is now indicated in the relevant figure panel as well as in the legend, for Figure 4 the passage number is indicated in the revised legend. Other figures now contain passage numbers as well, where appropriate.

2. *I’m not convinced using S4 fraction translates into better information compared to S3 fraction. In Figure 1c, the differential reads distribution for S4 vs S2 and S3 vs S2 seem to be quantitative rather than qualitative. Can the authors provide some analyses that clearly show the advantage of S4 over S3? How much more information a user of the technology gets when including S4 as well?*

We are aware that the S3 and S4 fractions have quantitative differences. However, we focused on the S4 vs S2 comparison, as it is the most consistent comparison, showing an average of 70.18% conservation (Supplementary Figure 1f). Moreover, we confirmed that lamins are in the S4 fraction by Western Blot (Supplementary Figure 1a).

Nevertheless, as suggested by the reviewer, we added results based on S3 vs S2 comparison to Figure 6a, Supplementary Figure 2a (related to main Figure 2b) and to Supplementary Figure 2c.

It's worth remarking that we already had included results based on both S4 vs S2 and S3 vs S2 in Figures 3b, 3c and Supplementary Figure 5a, 5b. Overall, the main differences between control and progeria samples were captured in the S4 vs S2 comparison.

3. Can the authors provide some mechanistic evidence for the early changes of chromatin structure in HGPS fibroblasts? For example, overexpression of progerin in control cells followed by SAMMY-seq analysis might recapitulate the profile observed in HGPS patients. Or would a knock-down of progerin in HGPS cells and cause some rescue in the SAMMY-seq profiles? This type of experiments would also widen the array of applications of the technology.

As suggested by the reviewer we considered different strategies to address this point. Previous studies showed that silencing of transcript variant 7 specifically depletes progerin transcript without affecting Lamin A/C transcripts (Piekarowicz et al., Cells 2019). Unfortunately, these experiments were performed in hTERT-immortalized cells (Huang et al., Hum. Genet. 2005) or in iPSCs (Liu et al., Cell Stem Cell 2011, Zhang et al., PNAS 2014), not directly comparable with our model system. We also reasoned that progerin knock-down by RNA interference would require subsequent cells replication and passaging to achieve the clearance of accumulated progerin, which is known to be more resistant to protein degradation within cells (Wu et al., Nucleus 2016). This would hamper the possibility to examine the knock-down in early passage HGPS cells.

The alternative approach based on progerin induction can be performed in the previously published human fibroblasts with a stably integrated doxycycline inducible construct expressing a GFP-progerin fusion gene (Kubben et al., Cell 2016). Of note, these cells are immortalized, thus not reproducing the cellular senescence phenotype of normal primary fibroblasts that we used instead in our study.

We decided to adopt the second experimental strategy. We performed three biological replicate experiments, considering four time points (time 0h/untreated, 24h, 48h and 96h post induction). Notably, we designed and performed the experiment with parallel treatments for the different time points, in order to collect and process all cell samples on the same day.

We first noticed that when examining the S4 vs S2 fractions comparison the untreated samples didn't show the expected SAMMY-seq pattern observed in normal primary fibroblasts. This suggested a difference in immortalized cell lines compared to primary fibroblasts. When examining the S3 vs S2 fractions comparison, we found instead that 2 out of 3 replicates in the untreated time 0 control showed a SAMMY-seq pattern partially concordant with results obtained in normal primary fibroblasts (see Figure R6 below).

At this stage, we can't confidently define if this variability is due to the different behaviour of immortalized cell lines in culture or to an experimental failure.

Figure R6: SAMMY-seq in healthy control samples and inducible model time zero. Genomic tracks for a representative region in chromosome 10 (coordinates and ideogram in the upper part of the figure) for SAMMY-seq differential enrichment for S3 vs S2 fractions comparison for (from top to bottom) the three healthy human fibroblasts used in the manuscript along 3 replicates for time zero of the inducible progerin model, based on immortalized fibroblasts. For each sample, the significantly enriched peaks are marked with a grey rectangle below the corresponding track with enrichment signal.

Interestingly, we noticed a dynamic change based on the 2 replicates with the expected S3 vs S2 pattern. Namely, we observed a specific change in the SAMMY-seq profile at 48 hours post induction, and a recovery of the control profile at 96 hours (see Figure R7 below). However, due to the aforementioned variability among replicates, we can't confidently draw a specific conclusion on this experiment and we decided not to include these data in the revised manuscript.

Figure R7: SAMMY-seq in the inducible model time course. Genomic tracks for a representative region in chromosome 10 (coordinates and ideogram in the upper part of the figure – same region as in figure R5 above) for SAMMY-seq differential enrichment for S3 vs S2 fractions comparison for two replicate time course experiments (time 0h, 24h, 48h and 96h for each replicate) of the inducible progerin model, based on immortalized fibroblasts. For each sample, the significantly enriched peaks are marked with a grey rectangle below the corresponding track with enrichment signal.

4. *PLA assay shows that Lamin A interacts with PcG proteins in control or HGPS fibroblasts whereas progerin interacts with Ezh2 in late passage HGPS. However, it would be more convincing if the authors would provide evidence also with a biochemical assay, showing the same interactions by using immunoprecipitation assay as well.*

As the reviewer suggested we indeed initially performed co-immunoprecipitation experiments to assess the interaction between progerin and Polycomb proteins. We used the only commercially-available, “Co-IP grade”, antibody for progerin, used in a previous study (Harhour et al., *Embo Mol Med* 2017) to detect interaction between Lamin A and progerin. This antibody, in our hands had low efficiency for immunoprecipitation assays. We are showing here results of co-immunoprecipitation of progerin, Ezh2 and Lamin A/C in healthy human fibroblasts (CTRL004 as used in the manuscript – passage number 27) and HGPS primary fibroblasts (HGPS167 – passage number 20) (see Figure R8 below). In this experiment we can see an interaction between Ezh2 and progerin only using Ezh2 antibody for the immunoprecipitation.

Figure R8: Western blot analysis of colP performed in healthy and HGPS primary fibroblasts. Nuclear extracts immunoprecipitated with lamin A/C, Ezh2 and progerin antibodies, together with inputs (labels on top), were immunoblotted and hybridized with indicated antibodies (labels on the left). An unrelated antibody (mouse IgG) was used as negative control.

However, we would remark that PLA assay is a commonly used method for in vivo analysis of protein-protein interactions, allowing the visualization and intracellular localization of such interactions (Fredriksson et al., *Nat. Biotechnol.* 2002). In general, PLA is considered a more sensible and specific technology, thus we relied on PLA results instead of the co-immunoprecipitation data (Smits and Vermeulen, *Trends Biotechnol.* 2016; Alam, *Curr Protoc Immunol.* 2018; Young, *Methods Mol Biol.* 2019).

5. *The authors correctly mentioned that Hi-C experiments failed to detect alterations for 3D DNA contacts in early passage HGPS as they were evident only at late passage cells. SAMMY-seq detects reduced enrichment of S4 vs S2 in HGPS fibroblasts only, suggesting alterations in chromatin organisation in patients cells. How about SAMMY-seq signal at late-passage HGPS? Does the S4 vs S2 enrichment profile change? Is it completely lost?*

We'd like to remark that in line with previous literature (McCord et al., *Genome Research* 2013; Zhang et al., *Plos One* 2016) early passage progeria patient primary cell cultures are considered an experimental model to study early molecular alterations in HGPS. Nevertheless, most of the previous literature on progeria is generally focused on later passage cells (Larrieu et al., *Science* 2014; Larrieu et al., *Sci Signal* 2018; Liu et al., *Nat Comm* 2013; Shumaker et al., *PNAS* 2006; Mattioli et al., *Aging Cell* 2018; Pellegrini et al., *Oncotarget* 2015) where morphological and functional alterations are more evident.

For these reasons we originally focused our experiments on early passage HGPS cells. It's worth remarking that at the passage numbers used in the original manuscript (between 10 and 14 passages) the HGPS fibroblasts show morphological features comparable to normal control fibroblasts.

As suggested by the reviewer, we maintained one of the three progeria primary cell lines (HGPS167) in culture up to passage 20 (about 3 months in culture from the early passage cells used in the previous manuscript version). We performed SAMMY-seq on 2 biological replicates (2 batches of cells independently passaged in culture). We found that S4 vs S2 fractions comparison yields a pattern suggesting a partial recovery of the control fibroblasts profile (Figure R9 below). This is shown in the genomics tracks (Figure R9a), in the pairwise correlation between SAMMY-seq profiles as well as in the overlap between SAMMY domains (Jaccard index scores) (Figure R9 c,d). On the other hand, in the S4 vs S3 comparison the control and HGPS profiles are still very different (Figure R9 b,e,f) and confirm the lost association with heterochromatin marks, as shown in the metaprofiles around SAMMY-seq domain borders (Figure R9 g,h).

Two considerations may explain the partial recovery of the control fibroblasts profile in the S4 vs S2 comparison in late passage HGPS cells:

- 1) It must be noted that progerin accumulation is known to block cell proliferation. Therefore, the long time in culture may actually counter-select the cells with more abundant progerin. Thus the high number of passages may enrich for cells closer to the normal cells phenotype also in the HGPS samples.
- 2) We should consider that SAMMY-seq does not rely on crosslinking. Thus the initial step of cell and nuclear membranes permeabilization may cause the premature rupture of the most senescent cells. This as well may contribute to selecting cells with a more “normal-like” phenotype in later passages.

In conclusion, we reasoned that these specific observations on late passage cells would require more extensive investigations that would be beyond the scope of the original manuscript entirely focusing on early passages. For these reasons we would prefer not to include these data in the revised manuscript, but we just included Figure R8 in the reply to reviewers.

Figure R9: SAMMY-seq profiles in early and late passage cells. **a)** Genomic tracks for SAMMY-seq profiles (S4 vs S2 fractions comparison) for a representative region in chromosome 5 (coordinates in the higher part of the plot) for healthy human fibroblasts (CTRL004) and HGPS primary fibroblasts (HGPS167) at early passage, as well as 2 biological replicates for each of them (R1 and R2) at later passage. **b)** same as panel (a) but for S4 vs S3 fractions comparison. **c)** Pairwise comparison (Kernel correlation) of the SAMMY-seq profiles for S4 vs S2 fractions comparison. **d)** Pairwise overlap (Jaccard index) of the SAMMY-seq domains called in the S4 vs S2 fractions comparison. **e)** same as panel (c) but for S4 vs S3 fractions comparison. **f)** same as panel (d) but for S4 vs S3 fractions comparison. **g)** metaprofile for the smoothed average ChIP-seq enrichment signal (y-axis) for Lamin A/C, Lamin B1 or H3K9me3 ChIP-seq (labels on the left side) around the start (left side metaprofiles) or the end (right side metaprofiles) of SAMMY-seq domains called in the S4 vs S2 fractions comparison for either early or late passage control (blue lines – colour legend) or progeria (red lines – colour legend) samples, using a +/-50 bins window (10Kb bin size) centred on the start or end domain border positions (see also grey boxes cartoon at the bottom). **h)** same as panel (g) but for S4 vs S3 fractions comparison.

6. *The authors do not find alterations in H3K9me3 patterns when performing ChIP-seq using antibody against this specific histone modification. One possible explanation for this result is the use of formaldehyde to crosslink cells that dramatically decreases to signal-to-noise ratio. Recently, a new technology, Cleavage Under Targets and Release Under Nuclease (CUT&RUN, Skene at al, Elife 2017), has been published combining native conditions and antibody-directed MNase digestion to ensure that chromatin cleavage happens solely close to protein of interest. This technology does not require the use of crosslinking agents or extensive sonication, thus greatly reducing the signal from unwanted genomic regions. The authors should perform CUT&RUN using H3K9me3 antibody to convincingly show the absence of significant differences in H3K9me3 patterns between controls and HGP patients.*

The data reported in our manuscript include transcription profiles (RNA-seq) for the same samples on which we performed H3K9me3 ChIP-seq. As expected, the genes located inside H3K9me3 domains show consistently low expression levels, as compared to the genes outside H3K9me3 domains (Supplementary Figure 6d). These results confirm that these domains are transcriptionally repressed. Thus the lack of changes in H3K9me3 is confirmed by these independent functional genomics data.

We are aware that CUT&RUN might improve the signal-to-noise ratio compared to ChIP-seq, as highlighted by Steven Henikoff's publications (Skene at al., Elife 2017) and (Meers et al., Elife 2019).

However, even if the improved resolution of CUT&RUN should highlight localized changes, still the global expression profile is not changed in the H3K9me3 domains as discussed above (Supplementary Figure 6d). Moreover, the current work is focused on identifying early aberrant molecular alterations in progeria. The analysis of transcription profiles showed that the changes in expression are more related to H3K27me3 and Polycomb regulation (Figure 5a), thus involving regulation mechanisms not related to H3K9me3. As such, we deemed CUT&RUN analysis beyond the scope of the current work.

Minor:

- *From the main text it is not clear which are the major differences between SAMMY-seq and previously published NGS methods that employ chromatin fractionation. Can the authors provide a table that compares across SAMMY-seq and other methods with regards to conditions, enzymes and cellular models? This would help both reviewers and readers to understand the advantages of SAMMY-seq over already existing NGS methods.*

As suggested by the reviewer, we summarized in a table the main characteristics and differences between SAMMY-seq and other genome-wide methods to study LADs and heterochromatin including ChIP-seq, DamID, gradient-seq (Becker *et al.*, Mol Cell 2017; Nicetto *et al.*, Science 2019) and protect-seq (Spracklin and Pradhan, Nucleic Acids Res. 2020). We included this table in the revised supplementary information (Supplementary Table 5).

- *I do not find surprising that correlation becomes more stable for genomic bins of larger size. Can the authors provide analyses to estimate the resolution of SAMMY-seq?*

We further refined the analysis of correlation between SAMMY-seq data by adding more small bin sizes (starting from 500bp) now presented in the revised Supplementary Figure 1e, where the x-axis reporting the bin size is now in logarithmic scale to improve visibility of results across several orders of magnitude. The results starting from this finer grain analysis confirm that the correlation tends to reach a plateau around 1Mb, for both S3 and S4 fractions. This is also in line with a) the order of magnitude of the SAMMY-seq domains size (see also Supplementary Figure 1h); b) the visual inspection of read coverage profiles showing megabase-scale "bumps" in S3 and, more prominently, in S4 fractions (Figure 1b); c) as well as the order of magnitude of LADs, according to literature (van Steensel and Belmont, Cell 2017).

- *The authors keep on changing the chromosomal regions shown in the Main Figures. What is the rationale for this as it seems rather confusing. I suggest to use same chromosomal regions or at least give the rationale why authors constantly change the chromosomal area.*

We decided to show different genomic regions so as to confirm that SAMMY-seq relevant patterns are not chromosome-specific or genomic region-specific. For example, the readers can appreciate that control samples have similar SAMMY-seq profiles in any genomic region that is shown, thus confirming we didn't "cherry-pick" a single nice looking region.

- *Can the authors speculate why S2 fraction is more variable across biological replicates? Is it somehow related to the digestion time of DNase?*

We confirm that the digestion time is always the same, as indicated in the Methods section. As illustrated in Figure 1b, the S2 fraction is generally showing a mostly flat reads distribution profile. As such S2 does not have a clear signal to noise separation and this is expected to give more variability.

- *Related to major point 2: it is needed to show also correlation for S3 vs S2, S4 vs S2 and S3 vs S4 to establish how different the fractions are with each other.*

As suggested by the reviewer, we performed the correlation analysis also on the other comparisons and included the updated results in the novel Supplementary Figure 2a, confirming that the results are mostly similar.

- *Also related to major point 2: it would be very useful to perform the same analyses depicted in Supplementary Figure 2c using S3 vs S2 fractions.*

As suggested by the reviewer we updated Supplementary Figure 2c by adding the overlap to the S3 vs S2 SAMMY-seq domains.

Reviewer #3 (Remarks to the Author):

In this manuscript, the authors developed a high-throughput sequencing-based method, SAMMY-seq to detect the genome-wide heterochromatin accessibilities in primary cells. With this approach, the authors first characterized the chromatin accessibility changes in early-passage fibroblasts derived from HGPS patients, and their correlations with dysregulated H3K27me3 marks and Polycomb bivalent genes expression. They drew the conclusion that “chromatin structural changes are early events in HGPS nuclear remodeling and interfere with proper PcG control”. This work provided new insights into understanding the early-onset heterochromatin remodeling events that contribute to premature senescence in HGPS fibroblasts. Outlined below are specific comments to help improve this manuscript.

Major Issues

- In this study, SAMMY-seq enrichment was normalized with signals from less condensed fractions. However, S4/S2 vs S4/S3 sometimes show different trends (e.g. Fig 3c). Under circumstances of inconsistency, is there a criterion to determine which normalization is more reliable?*

As for a similar question raised by reviewer #2, we confirm that we generally focused on the S4 vs S2 comparison, as it is the most consistent comparison, showing an average of 70.18% conservation (Supplementary Figure 1f). Moreover, we confirmed that lamins are in the S4 fraction by Western Blot (Supplementary Figure 1a).

However, the choice of S4 vs S2 as reference comparison was based on control samples. Instead, when considering HGPS samples we observe a generally higher variability in the SAMMY-seq profiles (Figure 3a). This large inter-individual variability between HGPS patients may also result in an intermediate profile as highlighted in Figure 3c, where 1 out of 3 HGPS samples show a different average pattern only for S4 vs S2 comparison. However, the S4 vs S3 comparisons clearly separate HGPS and control samples.

The availability of more fraction comparisons in SAMMY-seq allows capturing these more subtle variations.

We amended the manuscript main text as follows:

“Overall, these observations confirm a high inter-individual variability among HGPS patients and suggest that HGPS samples lose the pattern normally observed in controls.”

- Why are the SAMMY-seq regions preferentially overlap with the lamin A/C associated domains comparing to lamin B associated regions?*

As detailed in a similar comment by Reviewer #1, following this important point raised by the reviewers, we analysed several published ChIP-seq datasets for Lamin A/C and Lamin B1 in human fibroblasts. As detailed in the new Figure 2b, the average correlation with SAMMY-seq samples is similar for Lamin A/C and B1, despite some variability across individual samples (see new Supplementary Figure 2a). Using the novel results on the extended set of ChIP-seq datasets for Lamin A/C and Lamin B1 we updated Figures 2b and 2c, as well as Supplementary Figures 2a, 2b and 2c.

Based on these results, the inter-dataset variability in Lamin A/C or Lamin B1 does not allow us to make a specific statement about the differences between the two lamins.

- What are the passage numbers of the control fibroblasts? In this study, the patterns of SAMMY-seq domains showed decent consistency among different control samples, will this change over passages/aging?*

As explained for a similar point raised by other reviewers, we have now revised the manuscript and figures to make sure that the exact passage number is cited in each figure panel and legend.

We used higher passage numbers for control fibroblasts (around 20 passages) with respect to HGPS fibroblast (between 10 and 13 passages), as indicated in each figure. However, for another request of

reviewer #2 we maintained one of the three control primary cell lines (CTRL004) in culture up to passage 27, and then we performed SAMMY-seq on 2 biological replicates (2 batches of cells independently passaged in culture). We found that S4 vs S2 fractions comparison are less consistent in the pairwise correlation going from an average among early passage cells of 0.79 (Supplementary Figure 1d) to an average of 0.59 in comparison with late passage cells (Figure R10).

Figure R10: Pairwise comparison (Kernel correlation) of the SAMMY-seq profiles for S4 vs S2 fractions comparison in early and late passage control cells.

• *Figure 3a: the SAMMY-seq domains appeared scattered in different HGPS samples. Is it possible to find an HGPS-specific pattern?*

As indicated in the main text, the HGPS samples have a highly variable and not reproducible pattern (average Jaccard Index 0.11) as opposed to the control samples having a more reproducible set of SAMMY-seq domains (average Jaccard Index 0.67) (Figure 3b). Quoting from the manuscript text:

“SAMMY-seq domains (S4 vs S2) appeared more variable in number and dimension across samples with respect to controls (Supplementary Figure 1h-i), and scattered and with generally lower signal-to-noise ratio (Figure 3a) in HGPS, as opposed to the more consistent overlap across control replicates. This would be compatible with a general loss of chromatin organization following the lamina structural alteration, thus resulting in more random distribution of genomic regions among the different chromatin fractions. Indeed, SAMMY-seq domains are largely overlapping among control samples (average Jaccard Index 0.67) whereas SAMMY-seq domains are more variable in progeria samples, most notably in the S4 vs S2 comparison (average Jaccard Index 0.11) (Figure 3b and Supplementary Table 6)”

Thus, the peculiarity of HGPS samples is the loss of the normal (control cells) pattern, rather than the acquisition of new disease-specific pattern.

We have also remarked this point at the end of the same results paragraph as follows:

“Overall, these observations confirm a high inter-individual variability among HGPS patients and suggest that HGPS samples loose the pattern normally observed in controls.”

• *Figure 5b: HGPS188 is not significant in the selected region. Only 2 controls are shown.*

If the reviewer is implying that the regions should show a statistically significant difference in H3K27me3 peaks, it's worth remarking that we are not claiming that the region in figure 5b has a significant difference. Indeed, as stated in the text we acknowledge *“the main enrichment peaks seem to be present in both controls and HGPS samples”*.

If the reviewer is instead implying that HGPS188 is not showing a clear “spreading” of H3K27me3 enrichment in the selected region, we revised the figure to show a different region, in order to address also a related comment by Reviewer #4 (New Figure 5b). Along the same line, we now performed a quantitative evaluation on the spreading of H3K27me3 signal around annotated transcript start sites (TSS) (Supplementary Figure 8a-b) finding a less marked relative decrease in TSS distal vs proximal positions.

Then, in order to address also other related comments by Reviewer #4 we amended the figure by adding, the track for the location of SAMMY-domains detected in control and progeria samples and the track for gene annotations.

Finally, ChIP-seq for H3K27me3 was performed on three independent control samples for primary fibroblasts, but one of them failed. For this reason only two control H3K27me3 ChIP-seq are reported, as pointed out by this reviewer.

Minor Issues

- *Figure 1c: “Regions of signal enrichment or depletion over the reference samples are marked in yellow or purple, respectively” Where are the marks?*

We apologise for the inconsistency. We actually updated the figures colour scheme and the tracks showing the differential enrichment in the comparison between SAMMY-seq fractions are now coloured differently for each comparison. The description of SAMMY-seq fraction comparisons has now been updated in all figures.

- *Figure 4e: how was the quantification performed. What were the sample sizes?*

We apologise for the missing details. Indeed a portion of methods describing this quantification was erroneously removed in the last round of manuscript editing. We have now included a more detailed description in the Methods section:

“In order to compare these minimum distances among nuclei of different sizes we compute for each PcG body a measure of its closeness to nuclear periphery (proximity). This measure is size-independent, being computed by dividing the minimum distance by the distance of nuclear centroid from the point on nuclear periphery closest to the PcG body.”

Number of cell/total bodies/number of bodies close to the nuclear periphery (1 micron from nuclear periphery), respectively, are reported below:

CTRL001p22: 242 / 8598 / 434
CTRL002p17: 649 / 38160 / 2133
CTRL004p19: 518 / 34434 / 2006
HGPS164p11: 550 / 32885 / 1317
HGPS169p11: 65 / 2099 / 93

We included this information in the figure legend.

- Please fix typos

We carefully reviewed the text to amend typos.

Reviewer #4 (Remarks to the Author):

By sequential extraction of soluble fractions under different conditions, Sebestyén et al. have developed a method to extract heterochromatin of increasing compaction. By applying this method on progeria patient samples, authors have determined that chromatin organization changes take place at the most compacted heterochromatic regions (loss of such organization) that precedes H3K9me3 changes with earlier passages of primary cell culturing. They have also determined changes appear to take place at H3K27me3 enriched regions, specifically transcriptional repression at the bivalent regions appear to be lost.

General comments:

Although chromatin changes with the progeria patient samples are interesting, it felt like it took a long time to reach these observations. The first half that describes SAMMY-seq was not very easy to read through, due to points mentioned below, as well as the writing that should be improved to help the readers follow with more ease.

We carefully revised the text to improve readability. We marked all changes in red in the main text. We hope the reviewer will find the new version easier to read.

One thing I could not help but wonder is how different SAMMY-seq is with other non-antibody-dependent methods to extract heterochromatin such as by different salt concentrations Steve Henikoff had used. If these data are available for the same cell types, comparisons to other methods would improve the manuscript, especially if it can extract regions that are not by other already available methods.

In reply to a related comment by reviewer #2, we included an additional table to summarize main differences between SAMMY-seq and other methods to study heterochromatin structure (see Supplementary Table 5) and a table to more specifically compare the different conditions used by SAMMY-seq and other chromatin fractions based methods (see Supplementary Table 1).

The major difference between SAMMY-seq and chromatin fractions based methods is that our technique is the only one able to isolate the heterochromatin. In particular, Steve Henikoff “salt fractions profiling” method, which was mentioned by this reviewer, was mainly enriching for euchromatin (Henikoff et al., Genome Research 2009). In fact, Henikoff’s method has been generally used to examine open chromatin by his group as well as by other laboratories, across multiple models including articles on *Drosophila*, human, chicken and plant cells (Hu et al., PloS Pathogen 2019; Thakur et al., Genes and Dev. 2018; Gomez et al., Cell Reports 2016; Jahan et al., Epigenetics & Chromatin 2016; Sarg et al., J Proteomics 2015; Teves and Henikoff, Nat Struct Mol Biol 2014)

For what concerns the method details, “salt fractions profiling” was actually based on very different conditions compared to SAMMY-seq. Namely, Henikoff *et al.* used *Drosophila* cell lines, they digested chromatin by Micrococcal Nuclease, they separated fractions with different concentrations of salts, using different timing, without using urea and without intermediate wash steps (see Supplementary Table 1 for details).

More specifically, by carefully examining the article (Henikoff et al., Genome Research 2009) the readers can note that Figure 1b shows a Western blot of individual fractions, from which it is apparent that their pellet, obtained after salt extractions, still contains histones. Instead, our S4 fraction is free of histones (Supplementary Figure 1a).

Finally, to the best of our knowledge, there are no available datasets for “salt fractions profiling” in human primary fibroblasts.

Along this line, the novelty of this method is not very clear. If the most condensed heterochromatin is being extracted with this method which other methods do not, it should be more clearly stated. Although this aspect is mentioned in the Discussion, I do not see data in this manuscript that supports the statement ‘SAMMY-seq represents a significant improvement in the field of chromatin characterization

as it provides novel information complementary and not overlapping with other high-throughput sequencing based methods commonly used to study chromatin structure and function', since comparative analysis with other methods have not been carried out in this study.

Following the reviewer's suggestions, we deeply revised the text to improve clarity. More notably, we added two tables to more clearly illustrate the differences between SAMMY-seq and:

- 1) *Other genome-wide methods to study LADs and heterochromatin structure:* including ChIP-seq, DamID, gradient-seq (Becker *et al.*, Mol Cell 2017; Nicetto *et al.*, Science 2019) and protect-seq (Spracklin and Pradhan, Nucleic Acids Res. 2020) (Supplementary Table 5).
- 2) *Other genome-wide methods based on salt-extracted chromatin fractions:* including Salt fractions profiling (Henikoff *et al.*, Gen Res 2009) and HRS-seq (Baudement *et al.*, Gen Res 2018) (Supplementary Table 1).

As suggested below in another comment by this same reviewer, we moved to an earlier part of the introduction the comments on other techniques to improve clarity. Finally we performed a quantitative comparison analysis with Gradient-seq (Becker *et al.*, Mol. Cell 2017; Nicetto *et al.*, Science 2019) as discussed in another point by this same reviewer below (Figure R11).

Although the authors did not see differential gene expression at the protein coding genes, I wonder if expression of enhancer RNAs are affected at all. This may be outside the scope of this manuscript but something to consider for in the future.

Following this suggestion we indeed examined the expression of enhancer RNAs.

Enhancer regions definition. More specifically, we defined a list of putative enhancer regions based on the intersection of H3K27ac ChIP-seq and DNase-seq peaks calls in Roadmap Epigenomics data for human fibroblasts (E055 Roadmap sample ID). To make sure we considered distal enhancers we discarded peaks intersections located in a promoter proximal region (i.e. a window spanning from 3.5Kb upstream to 1.5Kb downstream of annotated transcription start sites - TSS). To avoid spurious signal from RNA-seq reads originating from nascent transcripts, we also discarded all peaks overlapping with gene bodies and converted the genomics coordinates from hg19 to hg38 using the UCSC liftover tool.

Quantification of enhancers transcription and differential expression. We calculated read counts for the enhancer regions defined above as well as for annotated transcripts. We had to quantify reads also for transcripts to reliably estimate variance and statistical significance for differential expression based on the limma tool and the voom method ($q\text{-value} \leq 0.05$). In the limma linear mode we took into account variables for sex, age and sequencing library as possible confounding factors.

We identified a total of 212 differentially expressed enhancers. We considered the closest TSS for each of these enhancers and we verified that there's limited overlap with the list of differentially expressed genes identified as described in the manuscript. We also considered the overlap between differentially expressed enhancers and the SAMMY-seq domains, but we identified only one differentially expressed enhancer being within the consensus regions for control samples for the S3 vs S2 fractions comparison.

These preliminary results did not suggest a specific connection between enhancers expression and SAMMY-seq data. We may need to further investigate more details about this potential connection to make a conclusive statement. However, given these preliminary observations and the reviewer remark that the analysis of enhancers expression may be beyond the scope of this article, we would not include these results in the revised manuscript.

I find the conclusions confusing at the end, since although changes are observed at the most condensed heterochromatin (S4 vs S2), the changes that are discussed are in the bivalent regions, which are not the most condensed heterochromatin. The logic or the model should be explained better.

Based on this and other related comments by reviewers we revised and extended the discussion of results to clarify the conclusions. Namely, for what concerns this specific point, we clarify that we

observe changes in chromatin distribution across nuclear compartments characterized by distinct accessibility, which are separated and detected by SAMMY-seq.

These changes in chromatin organization, and in particular the displacement of heterochromatin domains normally associated to the nuclear lamina (LADs) is not expected to yield large scale changes in silenced heterochromatin domains distribution along the genome, as shown also in recent literature (Falk et al., Nature 2019). We clarified in the text that the model we are proposing is in line with an alteration of the normal LADs interaction with lamina, but this is not yet causing a major alteration of H3K9me3 heterochromatin domains as confirmed by the persistent silenced state of these regions.

However, the alteration of lamina is affecting Polycomb regulated genes. Indeed, the alteration of LADs interaction with lamina is not expected to cause large gene expression changes *per se*, but it has been linked to alterations of Polycomb regulated genes also in other literature reports (Zheng et al., Mol Cell 2018). The crosstalk between Lamin A and Polycomb, firstly reported by us in (Cesarini et al., J Cell Biol 2015) is now supported by several independent studies (Zheng et al., Mol Cell 2018; Briand et al., Nucleus 2018; Oldenburg et al., J Cell Biol 2017; Salvarani et al., Nature Communications 2019; Bianchi et al., JCI 2020). The alteration of Polycomb regulation may be expected to affect first the so-called “bivalent” genes, which are already in an intermediate state, mostly silenced but primed for activation by the presence of both repressive (H3K27me3) and activating (H3K4me3) chromatin marks.

As such, our results are reasonable in light of all of the above-mentioned evidence in recent literature.

It would help the readers to have the histone modification antibody target name by the browser shots in the figures.

We thank the reviewer for noting these missing details. In fact the labels of some genomic tracks were accidentally omitted in a last round of figure revisions before submission. We apologize for the inconvenience and we confirm that all tracks in the genome browser-like images have been explicitly labelled in the revised manuscript.

Specific comments:

In the third line of the Results, it is written ‘DNase-sensitive chromatin (S2 fraction), ...’. Is this a typo for ‘insensitive’? Obviously the S2 fraction does not contain DNase-sensitive chromatin, evident from the browser shot profiles of the S2 fractions. If it is DNase-sensitive, it should look like DNase-seq and ATAC-seq, which it does not. I believe it contains the chromatin that was not released by DNase. It took some time for me to figure this out at the beginning and I was very confused reading the manuscript especially when looking at the figures. This needs to be addressed in the text. Also although it is described in the figures, some descriptions of how sequential extraction was carried out should be mentioned in the text such as using urea for S4 fractions.

We agree with the reviewer that this description may be misleading. The S2 fraction is indeed the supernatant obtained after DNase treatment, whereas the pellet resulting after centrifugation would be the really DNase “insensitive” material. For this reason we would prefer not to call S2 as “DNase-insensitive”. To avoid confusion, we revised the text to explicitly comment on this point and defined the S2 fraction as “DNase-treated chromatin” amending the text as follows:

“DNase-treated chromatin, i.e. the supernatant obtained after DNase treatment (S2 fraction)”

It’s also worth mentioning that the DNase digestion conditions applied here are different from those used in the DNase-seq protocol. Indeed, for our S2 fraction we digest DNA with Turbo DNase in CSK buffer for 60 minutes at 37°C, whereas for DNase-seq the digestion with DNase-I is performed for 10 minutes at 37°C (Boyle *et al.*, Cell 2008).

We realized this point may be confusing as suggested by the reviewer. Thus we amended the text by adding the following sentence:

“It’s worth remarking that the S2 profile is not comparable to standard DNase-seq profile due to substantial differences in the digestion conditions applied here (see Methods).”

We also describe the use of Urea:

“[...] and the most condensed and insoluble portion of chromatin, extracted with urea buffer that solubilizes the remaining proteins and membranes (S4 fraction)”

Page 6: 7th line of results: ‘We first applied SAMMY-seq to 3 independent normal skin primary fibroblast cell lines, ...’. Instead of using the word ‘normal’, ‘healthy’ should be used (e.g. 3 independent normal skin primary fibroblast cell lines, originating from 3 different healthy individuals’

We amended the text and all references to “normal” fibroblasts as suggested by the reviewer.

Page 6: ‘At 1Mb resolution, the mean Spearman correlation is 0.27 for the S2 fraction, 0.92 for S3, and 0.79 for S4.’: correlation between what? Controls? If so, why is S4 vs S2 more consistent? If S2 fraction is less reproducible, it’s not clear to me why S4 vs S2 would result in more conserved.

The sentence cited by the reviewer is commenting on Supplementary Figure 1e, which is indeed showing correlation between pairs of control samples. Thus we amended the text to explicitly mention this, as suggested by the reviewer:

“[...] the mean Spearman correlation between control samples is [...]”

The S2 fractions generally show either a flat profile, or a mostly flat profile with some “valley” of lower coverage, which corresponds to the S4 fraction “bumps” of signal enrichment (Figure 1b). As such, we can’t expect a high correlation value between S2 fractions across replicates, as they will mostly capture small coverage fluctuations. On the other hand, the comparison S4 vs S2 will always show enrichment bumps at the specific regions, as the S4 has a more evident enrichment pattern.

Page 7: ‘Of note, other chromatin fractionation-based NGS methods have been previously described 41,42, however they are not directly comparable to our technology as they adopt different conditions, enzymes and cellular models. ∴ this should be mentioned much earlier. In addition, other methods that detect heterochromatin by non-antibody-dependent approaches should be mentioned a little bit in more details than it is currently. For example, Becker et al., Mol. Cell 2017 and Nicetto, Science 2019.

As stated above in reply to another related comment by this same reviewer, we revised the text by adding two tables to highlight the differences between SAMMY-seq and other genome-wide methods to study LADs and heterochromatin structure (Supplementary Table 5) and other genome-wide methods based on salt-extracted chromatin fractions (Supplementary Table 1). Moreover, we moved to an earlier part of the introduction the comments on other techniques to improve clarity.

For what concerns specifically the Gradient-seq method described in (Becker et al., Mol. Cell 2017; Nicetto et al., Science 2019), we performed a quantitative comparison with results obtained with SAMMY-seq. Namely, we assessed the sensitivity in detecting H3K9me3 domains, by examining what percentage of H3K9me3 domains are detected either by SAMMY-seq or Gradient-seq in human fibroblasts. We used the corresponding H3K9me3 ChIP-seq data for each sample, thus using an equally favourable comparison for each one: i.e. H3K9me3 ChIP-seq for each of our control samples and the H3K9me3 ChIP-seq peaks from Supplementary Table 3 by (Becker et al., Mol. Cell 2017). As reported in (Figure R11) below, SAMMY-seq consistently identifies a larger percentage of H3K9me3 domains.

Figure R11: SAMMY-seq vs Gradient-seq sensitivity in detecting H3K9me3 domains. The barplot shows the percentage of H3K9me3 domains (y-axis) covered by SAMMY-seq domains in the indicated fractions comparison (S3 vs S2; S4 vs S2; S4 vs S3), as well as by Gradient-seq domains as defined in (Becker et al.).

Page 7: ‘Using data from the Roadmap Epigenomics consortium⁴³ and other publications^{29,44,45} we noticed that SAMMY-seq signal (S4 vs S2 enrichment) is highly consistent across biological replicates (CTRL002, CTRL004, CTRL013), ‘: I find this statement strange. Consistency between replicates do not requires other datasets (such as the Roadmap).

We agree with the reviewer that the original phrasing was confusing. We amended the sentence as follows:

“Using data from the Roadmap Epigenomics consortium and other public datasets we noticed that SAMMY-seq signal (S4 vs S2 enrichment) is inversely correlated with open chromatin marks (ATAC-seq and DNase-seq) and is positively correlated with Lamin A/C and B1 ChIP-seq signal (Figure 2a).”

Page 7: ‘Interestingly, the overlap with Lamin A/C associated domains was 21% higher than with Lamin B associated domains, suggesting that SAMMY-seq preferentially enriches for regions interacting with Lamin A/C. ‘: not sure if this can be stated, since it can be an antibody quality issue. Furthermore, if the used chromatin was prepared in a standard manner (as it is mentioned in the methods), the most condensed chromatin that is observed in the S4 fractions may not be contained in this ChIP experiment, therefore would not see these regions being immunoprecipitated.

We agree with the reviewer that antibody quality issues may influence the chromatin immunoprecipitation efficiency. Indeed, we mention that one of SAMMY-seq advantages is that it avoids the use of antibodies.

As detailed in a similar comment by Reviewer #1, following this important point raised by the reviewers, we analysed several published ChIP-seq datasets for Lamin A/C and Lamin B1 in human fibroblasts. As detailed in the new Supplementary Figure 2a, the average correlation with SAMMY-seq samples is similar for Lamin A/C and B1, despite some variability across individual samples. We summarized these results in the revised main Figures 2b by reporting the average across all of these datasets. We also updated the Figure 2c, Supplementary Figures 2b and 2c, by using a consensus set of Lamin A/C or Lamin B1 LADs: i.e. domains identified in at least 2 Lamin A/C or Lamin B1 ChIP-seq samples, respectively.

Based on these results, the inter-dataset variability in Lamin A/C or Lamin B1 does not allow us to make a specific statement about the differences between the two lamins.

Page 10: *'To further investigate PcG role in early passage HGPS fibroblasts, we examined the total amounts of PRC1-subunit Bmi1 and PRC2- subunit Ezh2 (Supplementary Figure 6a, b), ': should be mentioned in the text that this is at the protein level.*

We agree with the reviewer, and we revised the text as follows:

"To further investigate PcG role in early passage HGPS fibroblasts, we examined the total amounts of PRC1 (Bmi1 subunit) and PRC2 (Ezh2 subunit) at the protein level (Supplementary Figure 7a, b), as well as H3K27me3 (Supplementary Figure 7c, d)."

Page 11: *'By visual inspection we noticed that while the main peaks were present in both controls and HGPS samples, H3K27me3 signal was spread over flanking regions in HGPS (Figure 5b)': this should be quantitated (amount of the genome enriched by H3K27me3).*

We acknowledge that a quantification of this pattern was missing. In the revised manuscript we included a novel figure 5b with additional details (to address another comment by reviewers). This novel figure 5b shows a more evident pattern of H3K27me3 signal outside the main peaks. Moreover we added a quantification of this pattern as described in the amended text and reported in Supplementary Figure 8a and b:

"To quantify this pattern, we considered the annotated transcription start site (TSS) for protein coding genes where H3K27me3 was detected in controls. We examined the H3K27me3 average profile around these TSS regions, where we detected generally lower enrichment in HGPS samples, as well as a less marked relative decrease in TSS distal vs proximal positions, yet not statistically significant due to variability across patients (Supplementary Figure 8a and b). This observation resembles the findings of H3K27me3 spreading described in another laminopathy model¹⁸."

Page 11: *'These findings overall suggest that early chromatin remodelling has an impact on a subset of PcG targets, the bivalent genes, more susceptible to variations of PcG occupancy⁴⁹ (Figure 6d). ': number of bivalent genes that are upregulated should be mentioned.*

As suggested by the reviewer we have now explicitly indicated the number of upregulated bivalent genes in the revised manuscript. We also added a statistical test to assess the significance of the overlap between bivalent and HGPS deregulated genes. The text has been amended as follows:

"To evaluate if bivalent genes are especially affected in progeria, we performed a Fisher test: we found 76 bivalent out of 257 differentially expressed genes in HGPS (Fisher test p-value < 2.2e-16); or 39 bivalent out of 144 up-regulated genes in HGPS (Fisher test p-value 4.32e-12). We thus analysed the H3K27me3 profile of the subset of bivalent genes up-regulated in HGPS, where we observed a clear drop in H3K27me3 signal (Figure 6b)"

Page 11/12: *'supporting the previous observation that PcG regulated regions shift towards the insoluble S4 fraction in HGPS (Figure 6b). ': Not sure how this is concluded from 6b.*

We understand the reviewer's concern that this may be too stretched. Thus, we removed the text cited by the reviewer in the revised version of the "Results" section.

I am also confused as to how Progerin-PcG crosstalk hypothesis was formed from the data.

The crosstalk between Lamin A and Polycomb firstly reported by ourselves in (Cesarini et al., J Cell Biol 2015) is now supported by several independent studies (Zheng et al., Mol Cell 2018; Briand et al., Nucleus 2018; Oldenburg et al., J Cell Biol 2017; Salvarani et al., Nature Communications 2019; Bianchi et al., JCI 2020) which are all cited in the revised "Introduction" section. Thus, this previous literature constitutes the rationale for the hypothesis of crosstalk between Polycomb and the mutated Lamin A (progerin) as well.

Figure 2: all the browser shots are so zoomed out. I'd like to see how they look when it is zoomed in more, are they continuous for a long stretch like H3K9me3? At this resolution, it looks like H3K27me3 is overlapping with H3K4me1 and open regions (ATAC and DNase), which should not be unless they are bivalent regions. Also, H3K4me1 track should be directly below ATAC/DNase tracks, bundling the active mark with the open regions.

As requested by the reviewer, we show below (Figure R12) a “zoom in” version of the original Figure 2a. On a large scale SAMMY-seq domains are located in H3K9me3 regions, which seem devoid of both H3K27me3 and H3K4me1. However, as the reviewer expected, on a more local resolution H3K4me1 and H3K27me3 have different patterns, with H3K4me1 peaks located at positions of relatively lower H3K27me3 enrichment, and vice versa.

We added an explicit comment about this point in the revised version of the manuscript:

“However, this should not be interpreted as a co-localization of H3K27me3 and H3K4me1. They are simply concordant in being absent from H3K9me3 enriched SAMMY-seq domains on a large scale. Indeed, a closer inspection of histone mark profiles shows that H3K4me1 peaks are located at positions of relatively lower H3K27me3 enrichment, and vice versa (Figure 2a).”

As requested by the reviewer, we also changed the colours and added labels to clearly present sequencing data in the revised Figure 2a.

Figure R12: Chromatin marks and SAMMY-seq enrichment. The plot shows genomics tracks for chromatin marks (ATAC-seq, DNase-seq, Lamin A/C, Lamin B1, H3K27me3, K3K4me1, H3K9me3) along with SAMMY-seq (S4 vs S2 fractions comparison) for a representative region in chromosome 5. This is a zoomed in version of the region presented in the original Figure 2a, to allow a closer inspection of the ChIP-seq tracks for H3K27me3 and K3K4me1, which are indeed showing peaks in different positions, as expected.

Figure 3a: Does the amount/number of SAMMY-seq domains change in progeria samples? Is it simple loss or are they acquired at all elsewhere? From the browser shot in figure 3a, it looks like there is a global loss but there seems to be acquisition in surrounding regions. Of particular notice, HGPS188

seems to acquire a large region. This should be mentioned in the text and whether general acquisition is observed should be analyzed.

We'd like to remark the total number, size and coverage of SAMMY-seq domains for each sample was reported in Supplementary Table 2. From these data we can see that the average total length of S4 vs S2 SAMMY-seq domains is 541.7Mb for control samples, and 554.7Mb for HGPS samples. Thus we can't make a statement on a difference in the total size of domains. Instead, as shown in the manuscript, we can state that 1) the HGPS SAMMY-seq domains are more variable than in control samples (Figure 3b) and 2) that SAMMY-seq domains normally present in controls are not detected in HGPS samples (Figure 3c).

Moreover, to provide a clearer representation of Supplementary Table 2 data, we added HGPS samples to the plots of Supplementary Figure 1h and 1i.

In such analysis, it may also be interesting to determine whether the acquired SAMMY-seq regions are as large as the controls. Additionally, it would also be nice to see if chromatin accessibility changes at all with the loss of SAMMY (S4 vs S2) domains (such as with ATAC-seq).

It's worth remarking that according to literature ATAC-seq signal is generally correlated with transcription activation. However, as we showed in our results we don't see a change in H3K9me3 and transcriptional activity within SAMMY-seq control domains when examining the same regions in early passage HGPS fibroblasts. Thus, we don't expect major changes in transcription-related chromatin accessibility as it would be measured by ATAC-seq.

We deemed ATAC-seq experiments to be beyond the scope of our work as we focused on heterochromatin remodelling.

In Figure 3c, S4 vs S2, two patients clearly have different patterns from the control samples, however, one patient shows similar profiles as the control samples. The patient sample numbers should be shown here so that it can be compared with other data such as the browser shots.

As suggested by the reviewer, we revised the style of figures to make sure individual samples are distinguishable. Namely, we updated figure panels 3b, 3c, 4c, 5c and 6b, as well as Supplementary figure panels 5a, 5b and 8a.

Figure 5b: would be good to see the SAMMY-seq data as well in the browser shot to see if it coincides with the loss of SAMMY-seq regions.

In order to address this and other related comments by Reviewer #3 we amended the figure by adding, the track for the location of SAMMY-domains detected in control and progeria samples and the track for gene annotations.

We didn't claim a specific local match between loss of SAMMY-domains and H3K27me3 distribution. As clarified in the revised manuscript version, the crosstalk between lamina and Polycomb does not imply a direct action of Polycomb regulation on LADs.

Figure 6a: In terms of the story flow (the H3K27me3 story) and placement in the text (at the end of the paragraph that describes figure 5), this figure belongs to Figure 5, although 'spreading' and 'loss' are not very consistent. This figure is also missing the legend on the y-axis. The number of genes being analyzed here should also be mentioned in the text at least.

Following the reviewer's suggestions we revised the story flow. In the new manuscript version this figure has been replaced by one focusing only on bivalent genes. As such it pertains to the current Figure 6, which is focused on bivalent genes (see in particular Figure 6b).

Figure 6d: the text only mentions about SAMMY-seq domains, but the figure is of H3K27me3 ChIP. This is very confusing and hard to follow logics. Additionally, it would be good to have the H3K4me3 ChIP-seq track to show bivalency.

In order to avoid misunderstandings about this point, we rephrased the sentence about Figure 6d (which is now figure 6c) as follows:

“These findings overall suggest that early chromatin remodelling has an impact on a subset of PcG targets, i.e. the bivalent genes, more susceptible to variations of PcG occupancy as observed also at the level of H3K27me3 distribution by ChIP-seq (Figure 6c).”

As suggested by the reviewer, we also revised the figure by adding labels and H3K4me3 track for normal human fibroblasts, to confirm that the gene is bivalent.

Minor comments

- Figure 2a: it doesn't break the manuscript, but ATAC-seq and DNase-seq are redundant, both of which assay chromatin accessibility

We understand the reviewer's point, but we would prefer to keep both tracks to provide a more comprehensive picture.

- Figure 2a: 'H3' should be added before the K9me3 etc, consistent with other figures (and be accurate nomenclature-wise)

Figure 3a: 'K9me3' should be 'H3K9me3' like elsewhere

We thank the reviewer for noting these missing details. As stated above for a similar comment, the labels of some genomic tracks were accidentally omitted in a last round of figure revisions before submission. We apologize for the inconvenience and we confirm that all tracks in the genome browser-like images have been explicitly labelled in the revised manuscript.

- Figure 3a: colors of lamin A/C and lamin B should be changed so that it is easier for the readers to distinguish data between these ChIP and SAMMY-seq.

As suggested by the reviewer we changed the color of Lamin ChIP-seq tracks in Figure 2a and Figure 3a to a brighter shade of blue, to make it more different from the SAMMY-seq tracks.

- for all browser shots, if the antibody used to ChIP is shown on the left, it would be helpful for the readers

As stated above, we apologize for the inconvenience and we confirm that all tracks in the genome browser-like images have been explicitly labelled in the revised manuscript.

REVIEWER COMMENTS

Reviewer #1 (Remarks to the Author):

I am quite satisfied with the revision of this manuscript. All my concerns have been addressed with some changes in the data that now make more sense to me particularly re the differences before between the lamin A and lamin B1 overlaps which would have potentially changed my thoughts on how the lamina interacts had the reanalysis not removed the differences. I thought before that the method was sufficiently different from others and should be published so that my biggest issue was the number of places that I got confused about exactly what had been done, similarly to reviewer 4; however, now I think that the changes to the text greatly improve the clarity so that everyone should be able to follow the story. I also read through the other reviewer comments and rebuttal to see if there was anything I missed and am content that the different issues they raised such as about early and late models have been adequately addressed. The only way to get a better model for this at this point would be highly unethical to say the least. Thus, I now fully support publication.

Reviewer #2 (Remarks to the Author):

In this revised version of the manuscript, the authors have provided further analyses and generated additional results to improve the overall quality of the article. Although I overall applaud the effort and amount of work that has gone in these revision, I regretfully remain perplexed over some aspects of the manuscripts as below:

1)I respectfully disagree with the authors explanation regarding the partial recovery of the control fibroblasts profile for HGPS cells at late passages. How do the authors explain the discrepancy with the extensive disruption of chromatin structure detected by Hi-C technology at late passages? If there is a sort of counter-selection against HGPS profile why Hi-C shows signs of senescence while SAMMY seq does not? I think that inclusion of the SAMMY-seq data for later passages should be included in the manuscript.

2)As the authors mentioned to have used SAMMY-seq with other cell types, I wonder whether the S4 vs S2 comparison is generally the most robust metrics for the methodology as shown with control fibroblasts? The rationale for this question stems from the observation that S4 vs S3 comparison identified differences between control and HGPS fibroblasts at late passages. In other words, a future user of the technology needs to sequence S2, S3 and S4 fractions and perform pairwise comparisons to find out the most solid one? If so the authors should clearly state it in the manuscript.

3)This reviewer is still concerned about the wide applicability of using a significantly lower amount of input cells for SAMMY-seq. Specifically, how would the S4 vs S2 comparison look when using far less HGPS fibroblasts? Would it still be possible to call SAMMY-seq considering the lower signal-to-noise ratio already present using 4 million cells?

4)I am concerned about the images depicted in Fig. 6d and 6e. The DAPI signal looks strongly overexposed. Can the authors provide information regarding exposition time for that and PLA signals as well?

5)I still find the text quite convoluted with multiple redundancies. I recommend some trimming and overall restyling to improve the flow and readability.

Reviewer #3 (Remarks to the Author):

The authors have done solid work to address my concerns. Thanks.

Reviewer #4 (Remarks to the Author):

Authors have revised the manuscript very well by addressing most of the points and the text has improved dramatically. Therefore the manuscript is fit for publication.

We thank the reviewers for their positive and constructive feedback. We detail below the modifications to the manuscript to address the few remaining points raised by reviewer #2.

Reviewer #1 (Remarks to the Author):

I am quite satisfied with the revision of this manuscript. All my concerns have been addressed with some changes in the data that now make more sense to me particularly re the differences before between the lamin A and lamin B1 overlaps which would have potentially changed my thoughts on how the lamina interacts had the reanalysis not removed the differences. I thought before that the method was sufficiently different from others and should be published so that my biggest issue was the number of places that I got confused about exactly what had been done, similarly to reviewer 4; however, now I think that the changes to the text greatly improve the clarity so that everyone should be able to follow the story. I also read through the other reviewer comments and rebuttal to see if there was anything I missed and am content that the different issues they raised such as about early and late models have been adequately addressed. The only way to get a better model for this at this point would be highly unethical to say the least. Thus, I now fully support publication.

We thank the reviewer for the positive feedback.

Reviewer #2 (Remarks to the Author):

In this revised version of the manuscript, the authors have provided further analyses and generated additional results to improve the overall quality of the article. Although I overall applaud the effort and amount of work that has gone in these revision, I regretfully remain perplexed over some aspects of the manuscripts as below:

1) I respectfully disagree with the authors explanation regarding the partial recovery of the control fibroblasts profile for HGPS cells at late passages. How do the authors explain the discrepancy with the extensive disruption of chromatin structure detected by Hi-C technology at late passages? If there is a sort of counter-selection against HGPS profile why Hi-C shows signs of senescence while SAMMY seq does not? I think that inclusion of the SAMMY-seq data for later passages should be included in the manuscript.

As suggested by the reviewer we have now included the SAMMY-seq data for later passage Hutchinson Gilford Progeria Syndrome (HGPS) patient fibroblasts in the revised version of our manuscript as Supplementary Figure 5. These data were previously incorporated in the reply to reviewers.

For what concerns the other two questions explicitly listed by the reviewer as related to this point, we should remark that:

- 1) In the original article by McCord et al. (Genome Research 2013), Hi-C data for late passage HGPS were sequenced with low coverage. Indeed, according to the original supplementary information by McCord et al., the number of Hi-C read pairs remaining after the filtering steps was 10.7 million for late passage HGPS cells vs 17 million for early passage HGPS cells. This means about 60% difference in read counts for the two samples. The substantial difference in coverage between early and late passage HGPS may raise some concerns on the robustness of the differences in Hi-C between early and late passage. This concern is heightened by the fact that both samples have relatively low coverage, especially if compared to more recent Hi-C datasets where sometimes even a billion of reads are sequenced to improve analysis resolution (Pal et al. Biophysical Reviews 2019). In our data instead, we show reproducible results in two independent replicates grown in parallel (Supplementary Figure 5) to confirm the robustness of the observation.
- 2) In the original article by McCord et al., for Hi-C experiments “early passage” HGPS were at passage 17, whereas “late passage” HGPS were at passage 19. The reported large change of Hi-C profiles over just 2 passages in culture may raise some additional concern about the robustness of the observation. In our case, instead, we used passages “10-13” as “early” and passage 20 as “late” passage HGPS, as we were aiming to characterize really distant phenotypes. As already indicated in the previous reply to

reviewers, and now remarked in the amended manuscript, considering that progerin accumulation blocks cell proliferation, this prolonged time in culture may result in selecting sub-population of HGPS cells more apt to survival in culture, i.e. showing a more “normal” phenotype.

- 3) Late passage HGPS cells tend to enter apoptosis with higher frequency (Hamczyk M.R. et al., *Embo Mol Med* 2019; Messner M. et al., *PLoS ONE* 2018; Hilton B.A. et al., *FASEB J* 2017; Atchison L. et al., *Sci Rep* 2017; Mehta I.S. et al., *Biochem Soc Trans.* 2010; Bridger J.M. et al., *Exp Gerontol.* 2004). As the Hi-C protocol starts with cell fixation with formaldehyde, apoptotic and cytolytic cells would be fixed and captured as well in the experimental protocol. Instead, SAMMY-seq protocol in the initial step of cell and nuclear membranes permeabilization (S1) would result in disrupting apoptotic bodies and discard any of their content in the initial steps. We also consider also this as an advantage of SAMMY-seq, as apoptotic bodies will not interfere with the observed phenotype.

We added specific comments about these points in the revised manuscript as follows:

“We also performed SAMMY-seq on late passage control (CTRL004) and HGPS (HGPS167) primary fibroblasts maintained in culture up to passage 27 and 20, respectively, i.e. about 3 months from the early passage cells (Supplementary Figure 5). Independently passaged HGPS biological replicates showed a pattern still different from late passage control sample in the overlap of SAMMY domains (Supplementary Figure 5a, and b). Nevertheless, we noticed a partial recovery of the control pattern in the S4 vs S2 comparison, as confirmed by genome-wide correlation (Supplementary Figure 5c) and association with heterochromatin marks (Supplementary Figure 5d). Of note, the S4 vs S3 comparison still highlighted consistent differences between the control and HGPS profiles (Supplementary Figure 5e-h), suggesting that SAMMY fractions give rise to complementary information in chromatin analysis.

The seemingly partial recovery of the normal fibroblasts pattern in late-passage HGPS cells can be explained by the progerin-dependent inhibition of the cell cycle and DNA damage accumulation⁵⁶⁻⁵⁸, generally resulting in more apoptosis in late passage HGPS cells. Thus, the prolonged culture may actually counter-select the cells with more abundant progerin. Moreover, the permeabilization of cell and nuclear membranes in the initial step of SAMMY-seq protocol would result in disrupting apoptotic bodies and discard their content, thus further under-representing the more damaged cells. It is worth noting that other chromatin analysis methods starting with a fixation step (e.g. crosslinking) would not equally clear apoptotic bodies, thus confounding the results.”

2) As the authors mentioned to have used SAMMY-seq with other cell types, I wonder whether the S4 vs S2 comparison is generally the most robust metrics for the methodology as shown with control fibroblasts? The rationale for this question stems from the observation that S4 vs S3 comparison identified differences between control and HGPS fibroblasts at late passages. In other words, a future user of the technology needs to sequence S2, S3 and S4 fractions and perform pairwise comparisons to find out the most solid one? If so the authors should clearly state it in the manuscript.

We agree with the reviewer that S4 fraction is not the only informative one and all chromatin fractions should be sequenced to achieve a complete characterization of chromatin. However, it is worth remarking that the previous manuscript version did not suggest to sequence only 2 fractions.

This may be even more relevant for future users applying the technique to other conditions and cell types, as recent publications by other groups point out as well the fundamental chromatin structure differences across different cell types (Tan L. et al., *Science* 2018). As such, the availability of multiple chromatin fractions should be regarded as an advantage of SAMMY-seq, to capture multifaceted differences in chromatin.

As requested by the reviewer, we explicitly added a statement about this point in the revised manuscript, in the Discussion section revised as follows:

“ All of the three SAMMY fractions are then sequenced as they convey complementary information about chromatin structure, which may have fundamental differences across distinct cell types.”

3) This reviewer is still concerned about the wide applicability of using a significantly lower amount of input cells for SAMMY-seq. Specifically, how would the S4 vs S2 comparison look when using far less HGPS fibroblasts? Would it still be possible to call SAMMY-seq considering the lower signal-to-noise ratio already present using 4 million cells?

We should remark that in the manuscript we explicitly stated that SAMMY-seq domains in HGPS are "scattered and with generally lower signal-to-noise ratio [...], as opposed to the more consistent overlap across control replicates. This would be compatible with a general loss of chromatin organization following the lamina structural alteration, thus resulting in more random distribution of genomic regions among the different chromatin fractions."

The bottom line of our statement is that SAMMY-seq domains in HGPS are neither robust nor reproducible, as indeed it would be expected in a cell context where chromatin structure regulation is compromised. Thus, we don't expect a robust signal even with a smaller number of cells in HGPS.

Instead, when considering cells with a clear pattern of chromatin structure such as normal fibroblasts, we confirmed that a reproducible pattern can be detected also with 10k cells.

4) I am concerned about the images depicted in Fig. 6d and 6e. The DAPI signal looks strongly overexposed. Can the authors provide information regarding exposition time for that and PLA signals as well?

We acknowledge the reviewer's observation that DAPI signal is saturated in those images. However, it is worth remarking that DAPI staining was not used for quantitative analyses. Instead, it was only used to demarcate the nuclear contour.

As far as the PLA signal is concerned, the acquisition settings were tuned to avoid saturation in all samples, in order to allow correct digital image quantification. Importantly, gain voltage and laser power were set using as negative control the healthy control fibroblasts treated with the same primary and secondary antibodies. In these progerin-free cells only minimal fluorescence could be detected with these settings.

We have now added this information in the figure legend of the revised manuscript as follows:

"The PLA acquisition parameters were the same in control and HGPS cells for each interactor pair"

We realized that we did not specify in the method section the microscope used for PLA acquisition and this could have created some confusion. We thus revised the methods section indicating the type of microscope used (confocal SP5 or Nikon ECLIPSE 90) for each experiment. In particular for PLA experiment we used the confocal SP5, in which the exposition time cannot be modulated. Instead, the variable parameters as the laser wavelength, the total laser power and percentage are now reported in the methods section.

For other figures in the manuscript we shared raw Western blot images and raw numeric data as per the guidelines related to sharing the "source data" of images. However, we could not find specific indications about how to share large scale raw imaging data in Nature Communications policies and procedures. All original PLA microscope images are available and can be provided as additional raw data if it is appropriate for the journal policies.

5) I still find the text quite convoluted with multiple redundancies. I recommend some trimming and overall restyling to improve the flow and readability.

We further revised the text to remove some sentences that were partially repeated in the introduction and initial part of the results sections. We also further revised the results sections to improve the phrasing of specific sentences containing many details and we added some statement to summarize the message of individual results paragraphs. Changes are marked in red in the text.

We note that reviewer #1 and #4 explicitly suggested a revision of the manuscript style in the first round of review. Now they both acknowledge that the text has been improved and easy to follow:

- reviewer #1: *"the changes to the text greatly improve the clarity so that everyone should be able to follow the story"*
- reviewer #4: *"the text has improved dramatically"*

We hope the additional revisions will be deemed appropriate.

Reviewer #3 (Remarks to the Author):

The authors have done solid work to address my concerns. Thanks.

We thank the reviewer for the positive feedback.

Reviewer #4 (Remarks to the Author):

Authors have revised the manuscript very well by addressing most of the points and the text has improved dramatically. Therefore the manuscript is fit for publication.

We thank the reviewer for the positive feedback.

REVIEWERS' COMMENTS

Reviewer #2 (Remarks to the Author):

I am satisfied with the revision of the manuscript. The authors have addressed all my remaining concerns.

We thank the reviewer for him/her positive feedback.

Reviewer #2 (Remarks to the Author):

I am satisfied with the revision of the manuscript. The authors have addressed all my remaining concerns.